# Evaluation of Liquid Cloud Albedo Susceptibility in E3SM Using Coupled Eastern North Atlantic Surface and Satellite Retrievals

Adam C. Varble[1], Po-Lun Ma[1], Matt W. Christensen[1], Johannes Mülmenstädt[1], Shuaiqi Tang[1], and Jerome Fast[1]

[1]Atmospheric and Global Change Division, Pacific Northwest National Laboratory, Richland, Washington, 99354, USA

*Correspondence to*: Adam C. Varble (adam.varble@pnnl.gov)

**Abstract.** The impact of aerosol number concentration on cloud albedo is a persistent source of spread in global climate predictions due to multi-scale, interactive atmospheric processes that remain difficult to quantify. We use 5 years of geostationary satellite and surface retrievals at the U.S. Department of Energy (DOE) Atmospheric Radiation Measurement (ARM) Eastern North Atlantic

(ENA) site in the Azores to evaluate the representation of liquid cloud albedo susceptibility for overcast cloud scenes in DOE Energy Exascale Earth System Model version 1 (E3SMv1) and provide possible reasons for model-observation discrepancies.

The overall distribution of surface 0.2% CCN concentration values is reasonably simulated but simulated liquid water path (LWP) is lower than observed and layer-mean droplet concentration ($N_d$) comparisons are highly variable depending on the $N_d$ retrieval technique. E3SMv1's cloud albedo is greater than observed for given LWP and $N_d$ values due to a lesser cloud effective

radius than observed. However, the simulated albedo response to $N_d$ is suppressed due to a solar zenith angle (SZA)-$N_d$ correlation created by the seasonal cycle that is not observed. Controlling for this effect by examining the cloud optical depth (COD) shows that E3SMv1's COD response to CCN concentration is greater than observed. For surface-based retrievals, this is only true after controlling for cloud adiabaticity because E3SMv1's adiabaticities are much lower than observed. Assuming a constant adiabaticity in surface retrievals as done in TOA retrievals narrows the retrieved $\ln N_d$ distribution, which increases the cloud albedo sensitivity

to $\ln N_d$ to match the TOA sensitivity.

The greater sensitivity of COD to CCN is caused by a greater Twomey effect in which the sensitivity of $N_d$ to CCN is greater than observed for TOA-retrieved $N_d$, and once model-observation cloud adiabaticity differences are removed, this is also true for surface-retrieved $N_d$. The LWP response to $N_d$ in E3SMv1 is overall negative as observed. Despite reproducing the observed LWP-$N_d$ relationship, observed clouds become much more adiabatic as $N_d$ increases while E3SMv1 clouds do not, associated with more

heavily precipitating clouds that are partially but not completely caused by deeper clouds and weaker inversions in E3SMv1. These cloud property differences indicate that the negative LWP-$N_d$ relationship is likely not caused by the same mechanisms in E3SMv1 and observations. The negative simulated LWP response also fails to mute the excessively strong Twomey effect, highlighting potentially important confounding factor effects that likely render the LWP-$N_d$ relationship non-causal. $N_d$ retrieval scales and assumptions, particularly related to cloud adiabaticity, contribute to substantial spreads in the model-observation comparisons,

though enough consistency exists to suggest that aerosol activation, drizzle, and entrainment processes are critical areas to focus E3SMv1 development for improving the fidelity of aerosol-cloud interactions in E3SM.

## 1 Introduction

Aerosol effects on liquid clouds are a longstanding leading source of uncertainty in climate projections (Boucher et al. 2013, Bellouin et al. 2020, Smith et al. 2020). As aerosols acting as cloud condensation nuclei (CCN) increase, the number of cloud

droplets ($N_d$) increases while droplet size decreases if holding other cloud properties constant. This increases the cloud albedo and is known as the first indirect, cloud albedo, or Twomey effect (Twomey 1974, 1977, Coakley et al. 1987, Radke et al. 1989).

However, this near-instantaneous aerosol-cloud interaction (ACI) can be buffered or amplified by slower cloud adjustments to the new cloud state including changes in cloud fraction (CF) and/or liquid water path (LWP) (Gryspeerdt et al. 2016, 2019, Christensen et al. 2020, and references therein). Increasing $N_d$ can increase cloud lifetime and thus CF and/or LWP by suppressing precipitation

that removes liquid from the cloud (Albrecht 1989, Pincus and Baker 1994), which is known as the second indirect, cloud lifetime, or Albrecht effect. However, it can also potentially decrease CF and/or LWP in non-precipitating clouds when dry air entrainment driven evaporation increases. This can occur via decreased cloud droplet sedimentation with increased cloud top radiative and evaporational cooling in stratocumulus clouds (Ackerman et al. 2004, Bretherton et al. 2007). Increasing $N_d$ may decrease cloud updraft equilibrium supersaturation, thus increasing condensate (Kogan and Martin 2004, Koren et al. 2014), but smaller droplets

may also increase cloud edge evaporation, thus decreasing condensate (Jiang et al. 2006, Xue and Feingold 2006, Small et al. 2009). There can also be several confounding factors affecting correlations between LWP and $N_d$ including meteorological correlations (Zhang et al. 2022). These individual factors influencing effective radiative forcing caused by ACI (ERFaci) are controlled by complex, cross-scale interactions between evolving clouds and their surrounding environment. Better isolating and quantifying these interactions and the processes that control them is required for reducing uncertainty and improving model

parameterizations.

The change in cloud albedo for a given change in $N_d$ or aerosols is called the cloud albedo susceptibility (Platnick and Twomey 1994). This metric is commonly decomposed into contributions from the first and second indirect effect where the second indirect effect can be further separated into changes in CF versus changes in LWP (e.g., Quaas et al. 2008, Mülmenstädt et al. 2019, Bellouin et al. 2020). These terms can then be further decomposed to isolate the response of $N_d$ to a change in aerosols, which we

refer to as the aerosol activation term herein while acknowledging that $N_d$ sink processes such as precipitation and evaporation also influence the $N_d$ vs. CCN relationship. Because cloud base CCN number concentration is rarely directly observed and very difficult to accurately retrieve from satellite measurements (Shinozuka et al. 2015), aerosol optical depth (AOD) or aerosol index (AI) are often used instead with the caveat that they can only be retrieved for clear sky and are thus offset in space from clouds (Stier 2016). These terms have been quantified in many observational and large eddy simulation (LES) studies that have elucidated

complex dependencies on cloud properties and atmospheric state that can yield susceptibilities of varying sign and magnitude (Zhang et al. 2022). The spread in estimated global Twomey effects remains substantial due to the need for imperfect aerosol proxies, $N_d$ retrievals, and statistical techniques used for observational quantification (Quaas et al. 2020), imperfect model parameterizations (Gryspeerdt et al. 2017), and an unknown change in aerosols between preindustrial and present day (Carslaw et al. 2013, Ghan et al. 2016). CF and LWP adjustments are even more uncertain given their operation over longer timescales and

larger spatial scales. The LWP response to $N_d$ in liquid clouds has been shown in observations and LES to be positive or negative depending on precipitation rate and evaporation via sedimentation-radiation-entrainment feedbacks (e.g., Lebsock et al. 2008, Chen et al. 2014, Glassmeier et al. 2019, Hoffman et al. 2020). The most recent net global $\frac{dlnLWP}{dlnN_d}$ estimates are negative (-0.3 to 0) with a positive value in precipitating clouds with $N_d <$ ~30 cm$^{-3}$ and a value as low as -0.4 in non-precipitating clouds with higher $N_d$ values (Bellouin et al. 2019). However, estimates vary even more substantially and go strongly positive on smaller scales

in which strong aerosol perturbations exist (Christensen et al. 2022). These values remain highly uncertain due to observational uncertainty and substantial modulation by atmospheric conditions such as inversion strength, relative humidity, and boundary layer depth (e.g., Gryspeerdt et al. 2019, Possner et al. 2020).

Present-day aerosol and cloud statistics are commonly used to quantify cloud albedo susceptibility or ERFaci, which are then used to evaluate ACI in climate models. Several previous studies have compared cloud albedo susceptibility, ERFaci, or similar

metrics in climate models with observations, often finding that models have a greater susceptibility than observed (Boucher et al. 2013, Ghan et al. 2016). More weight has been traditionally given to observational estimates, but recent studies have shown that

they can have significant biases, particularly depending on the aerosol variable used, with AOD being problematic (Penner et al. 2011), particularly near cloud edges (Christensen et al. 2017). Gryspeerdt et al. (2020) show that this can be largely attributed to the forcing being computed differently in observational and model datasets, and that climate models and observations have surprisingly good agreement if consistent methods are applied to each, though still with significant spread and disagreement on the sign and magnitude of cloud adjustments. In contrast to recent observational and LES studies showing a net decrease in global LWP in response to a $N_d$ increase, climate models most commonly produce a net LWP increase (Quaas et al. 2009, Gryspeerdt et al. 2020). This may be due to model representation of precipitation suppression with increasing $N_d$ but not buffering effects (e.g., Stevens and Feingold 2009) associated with condensate sedimentation, radiative cooling, entrainment, and evaporation (Ghan et al. 2016, Michibata et al. 2016, Toll et al. 2017). Indeed, there are suggestions that global storm-resolving models may better represent LWP responses to $N_d$ perturbations (Sato et al. 2018, Terai et al. 2020).

A difficulty in evaluating process parameterizations with observations is that such processes cannot be directly observed and need to be inferred from properties. A commonly used approach for analyzing processes given only statistics of select properties is plotting joint distributions and heatmaps of variables known to control and/or respond to processes (e.g., Suzuki et al. 2010, 2013, 2015, Gryspeerdt et al. 2016, 2017, 2019, Jing and Suzuki 2018, Zhang et al. 2022). Others have examined cloud susceptibilities to aerosols in the context of cloud and atmospheric state properties that modulate their magnitude (e.g., Douglas and L'Ecuyer 2019, 2020, Zhang et al. 2022). These approaches coupled with linear regressions to quantify relationships following numerous past studies are used in this study to assess how well the U.S. Department of Energy's Energy Exascale Earth System Model version 1 (E3SMv1; Golaz et al. 2019, Rasch et al. 2019) with ~1° grid spacing reproduces observationally estimated cloud albedo susceptibility controls at the Atmospheric Radiation Measurement (ARM) Eastern North Atlantic (ENA) site (Mather and Voyles 2013) and to identify possible reasons for discrepancies with observations.

Observational and modeling datasets as well as the study methodology are described in section 2. Comparisons of observed and simulated Twomey, LWP susceptibility, and aerosol activation contributions to cloud albedo susceptibility over the Eastern North Atlantic are discussed in sections 3-4 along with possible reasons for E3SM-observation differences. Finally, conclusions are presented in section 5.

## 2 Datasets and Methodology

### 2.1 Approach

Several methodological decisions are made to improve the interpretability of model-observation comparisons in this study. Both surface and satellite observational retrievals of cloud properties are used in comparisons with model output because of uncertainties associated with each, for $N_d$ in particular (Grosvenor et al. 2018). Although in situ cloud droplet measurements are more accurate than remote sensing retrievals, they are much rarer, which results in sampling biases. Surface and satellite retrievals are also applied to model output to yield multiple model datasets. Observational retrievals are analyzed at their highest resolution (5 min and 4 km) as well as coarse resolutions (60 min and 1°) consistent with model output. This approach yields 4 observational and 3 E3SMv1 datasets described in more detail below and in Table 1, where the output spread among them can be viewed as a rough measure of uncertainty stemming from retrieval assumptions and resolution. Lastly, analyses are confined to situations with single layer liquid clouds and greater than 95% cloud fraction to improve the accuracy of remote sensing retrievals (e.g., see Grosvenor et al. 2018).

Long-term surface retrievals require a fixed site that experiences frequent marine liquid clouds with aerosol, cloud, and atmospheric state measurements. They have several advantages over satellite retrievals including CCN observations that eliminate the need to use column-integrated optical properties such as AOD or AI, and which are co-located with clouds rather than offset

in space. In addition, no cloud adiabaticity assumption is required for retrieving $N_d$. Instead, adiabaticity can be estimated, and variables such as cloud depth and rain rate can be well quantified. This approach lends itself to Eulerian, climatological statistics as opposed to Lagrangian trajectory analyses (e.g., Pincus et al. 1997, Johnson et al. 2000, Eastman et al. 2016, Goren et al. 2019, Mohrmann et al. 2019, Christensen et al. 2020, 2023) or comparisons of situational aerosol plume effects on clouds from ship tracks, volcanoes, or other significant emission sources (see Christensen et al. (2022) and references therein), though those strategies provide valuable, complementary perspectives.

## 2.2 Observations

Surface datasets from 2016-2020 are collected from the ARM ENA site located on Graciosa Island in the Azores at about 39°N and 28°W (Mather and Voyles 2013). The site experiences a range of marine, mid-latitude meteorological and cloud conditions (Dong et al. 2014, Rémillard and Tselioudis 2015) with frequent drizzling stratocumulus (Rémillard et al. 2012, Giangrande et al. 2019, Jensen et al. 2021). Synoptic conditions strongly modulate clouds at ENA with substantial interannual variability, seasonal shifts in the Azores high and Icelandic low (Wood et al. 2015, Mechem et al. 2018), and frequent cold frontal passages (Naud et al. 2018, Kazemirad and Miller 2020, Lamer et al. 2020, Ilotoviz et al. 2021). Mesoscale moisture variability in the boundary layer has also been shown to be important (Cadeddu et al. 2023), while interactions between cloud top radiative cooling, drizzle evaporation, downdrafts, and boundary layer turbulence also affect the evolution of cloud properties (Ghate and Cadeddu 2019, Ghate et al. 2021). In addition, cloud properties at ENA experience a diurnal cycle with cloud deepening and drizzle production overnight (Rémillard et al. 2012, Dong et al. 2014, Ghate et al. 2021). Such processes have been shown to modulate the effects of aerosols on cloud microphysical and radiative properties (Zheng et al. 2022, Qiu et al. 2023). Despite the importance of this meteorological regime for climate prediction, few long-term surface-based and in situ measurements targeting aerosol, cloud, and radiation conditions exist for it outside of the ENA site, making it ideal for this study of aerosol effects on single layer liquid clouds (Wood et al. 2015) and for assessing findings based on satellite remote sensing retrievals.

Surface CCN concentration at 0.2% supersaturation is estimated by a CCN counter (ARM user facility 2016a) that varies supersaturation setpoints between 0 and 1% over the course of an hour from which a polynomial is fit to the data to provide the CCN spectra as a function of supersaturation (ARM user facility 2016b). Although aerosol measurements at ENA are occasionally impacted by emissions from the airport where the site is located (Gallo et al. 2020), this had a minimal impact on hourly 0.2% CCN statistics and thus we do not control for it. However, there are missing CCN measurements before 23 June 2016, between 31 July and 4 December 2018, after 28 October 2020. Thus, analyses involving CCN cover a shorter period than the full 5 years. Cloud LWP is retrieved from a 3-channel microwave radiometer (Turner et al. 2007, Cadeddu et al. 2013, ARM user facility 2014c) with an estimated uncertainty of ~20 g m$^{-2}$ for low LWPs increasing to ~10% for large LWPs (Cadeddu et al. 2013). A multifrequency shadowband radiometer (MFRSR) is used to retrieve cloud optical depth (COD) and layer-mean effective radius ($R_{eff}$) (Min and Harrison 1996, Min et al. 2003, ARM user facility 2014a). Times for which the $R_{eff}$ retrieval fails have a default constant $R_{eff}$ value, and these times are removed from analyses. Cloud base and top heights are retrieved from the Active Remote Sensing of Clouds (ARSCL) product (Clothiaux et al. 2000, ARM user facility 2015) that combines vertically pointing ceilometer and Ka-band zenith radar (KAZR) measurements. Cloud base and top temperatures are then estimated by matching temperature profiles retrieved from interpolated radiosonde measurements scaled by microwave radiometer precipitable water retrievals (Fairless et al. 2021, ARM user facility 2013a) to cloud base and top heights. Surface rain rates are retrieved using a Parsivel-2 disdrometer (ARM user facility 2014b) and optical rain gauge (ARM user facility 2013d). All-sky and clear sky downwelling broadband shortwave irradiances are obtained from a pyranometer via the RADFLUX product (Long and Ackerman 2000, ARM user facility 2013c), from which cloud effective albedo is computed by dividing the downwelling broadband shortwave flux by the

estimated clear sky downwelling broadband shortwave flux. These variables are then either averaged or interpolated to 5-min and 60-min intervals depending on whether the temporal frequency of the variable is less than or greater than that interval. $N_d$ is then derived using MWR-derived LWP and MFRSR-derived COD as in McComiskey et al. (2009) with Equation 1:

$$N_d = \frac{2^{-\frac{5}{2}}}{kH} \left(\frac{4\pi\rho_{liq}}{3LWP}\right)^2 \left(\frac{5COD}{3\pi Q}\right)^3 ,$$
(1)

where $H$ is the cloud depth, $k$ is the ratio of the drop volume mean radius to $R_{eff}$, $Q$ is the droplet scattering efficiency, and $\rho_{liq}$ is the liquid water density. This retrieval assumes a stratified adiabatic cloud model as in Bennartz (2007) but allows for variable cloud adiabaticity by incorporating $H$ following Boers and Mitchell (1994). LWP, COD, and $H$ inputs from surface retrievals are averaged to 5- and 60-min intervals for each of the surface observation datasets ("Obs Sfc 5min" and "Obs Sfc 60min" in Table 1). While it is more physically realistic to compute $N_d$ at the highest resolution possible and average it to coarser scales, that would be inconsistent with the E3SM-simulated surface retrievals that use inputs from a 1° grid. We assume $Q = 2$ following Bennartz (2007) and $k = 0.74$ following Brenguier et al. (2011). Values for $k$ commonly varies between 0.5 and 0.9 depending on the drop size distribution, which introduces uncertainty into the $N_d$ retrieval. Adiabatic LWP (LWP$_{ad}$) is estimated by moist adiabatic ascent of a parcel with the retrieved cloud base temperature and pressure to the radar-retrieved cloud top, from which cloud adiabaticity is computed as $\alpha = \frac{LWP}{LWP_{ad}}$. Lastly, CF is estimated from the frequency of clouds overhead within 5- and 60-min periods as detected in the ARSCL product.

Hourly satellite-based cloud retrievals are obtained from the NASA Visible Infrared Solar-infrared Split-Window Technique (VISST) dataset (Minnis 2008, 2011; ARM user facility 2013b, 2014c, 2018a-b). These retrievals use MeteoSat 10 and 11 geostationary satellite measurements to estimate top-of-atmosphere (TOA) radiative fluxes including all-sky albedo, cloud LWP and IWP, COD, and cloud top $R_{eff}$, height, and temperature for 4-km wide pixels and 0.5-deg regions, with the 0.5-deg retrievals including CF. To match the E3SMv1 1° grid, the 0.5° retrievals are averaged to 1° grids with each 0.5° value weighted by CF for CF-dependent variables to yield in-cloud rather than all-sky values. From these, cloud layer mean $N_d$ values for 4-km and 1° grids in the "Obs TOA 4km" and "Obs TOA 1deg" datasets (Table 1) are retrieved using Equation 2 following Bennartz (2007):

$$N_d = \frac{\sqrt{\alpha\Gamma_{ad}}}{k} \left(\frac{4\pi\rho_{liq}}{3}\right)^2 \left(\frac{5COD}{3\pi Q}\right)^3 (2LWP)^{-\frac{5}{2}} ,$$
(2)

where $\alpha$ is cloud adiabaticity (assumed to be 0.8 in this study) and $\Gamma_{ad}$ is the adiabatic liquid water content lapse rate. Equation 2 assumes adiabatically stratified clouds and uses inputs of COD, LWP, and cloud top temperature (for calculation of $\Gamma_{ad}$) that are obtained from the VISST retrievals. Because $\alpha$ is assumed to be constant, this retrieval may be less accurate than the surface-based retrieval using equation 1, though the accuracies of COD and LWP inputs also matter. Bennartz (2007) showed that the uncertainty in this retrieval is less than 80% for LWP exceeding 30 g m$^{-2}$ and CFs exceeding 0.8. Cloud adiabaticity is highly variable (e.g., Merk et al. 2016), which along with the assumed $k$ value of 0.74, cloud inhomogeneity, LWP uncertainty, and satellite viewing angle, contributes to $N_d$ retrieval uncertainty. Grosvenor et al. (2018) reported $N_d$ retrieval uncertainties of 54-78% depending on the averaging area for ideal conditions, while Gryspeerdt et al. (2022) reports lesser uncertainties of 30-50% for overcast stratocumulus with parameter thresholds to remove likely biased samples, most of which we employ in this study (see Section 2.4). Surface-measured CCN is interpolated to satellite retrieval times for analyses relating satellite-retrieved cloud properties with CCN concentration. An example of a closed cell stratocumulus case is shown to highlight some of the key retrieved variables from the surface and satellite data (Fig. 1).

## 2.3 Model Output

The E3SMv1 (Golaz et al. 2019) Atmosphere Model (Rasch et al. 2019) is run with ne30np4 horizontal resolution (approximately 1° grid spacing) for 2016-2020 with hourly output. Although E3SMv2 was recently released (Golaz et al. 2022), E3SMv1 is used because it has been better characterized to date in many studies and as part of the Coupled Model Intercomparison Project Phase 6 (CMIP6; Eyring et al. 2016). Large-scale meteorological conditions are constrained via nudging the horizontal winds toward the Modern-Era Retrospective Analysis for Research and Applications version 2 (MERRA-2) (Gelaro et al. 2017) with a relaxation

timescale of 6 hours. Several of the variables used in analyses are directly outputted including grid-scale LWP, CF by height, surface and TOA radiative fluxes, temperature and moisture by height, surface rain rate, and surface CCN concentration at 0.2% supersaturation.

Other variables require derivation. TOA albedo is computed as the upwelling broadband shortwave radiative flux divided by the downwelling broadband shortwave radiative flux at TOA. Cloud effective albedo is computed as the downwelling broadband

shortwave radiative flux divided by the clear sky broadband shortwave radiative flux at the surface. Cloud base height is derived by summing height levels at which CF is greater than 0 weighted by that level's CF contribution minus the integrated CF below that level to the total 2D low level CF moving upward from the surface until the 2D low level cloud fraction is reached. This assumes maximum cloud overlap. Mathematically, this is written as:

$$Cloud\ Base\ Height = \sum_{z=1}^{nlev} \left\{ Z_z max \left[ \frac{CF_z - max(CF_1, ..., CF_{k-1})}{max(CF_1, ..., CF_{nlev})}, 0 \right] \right\},\qquad(3)$$

where $z$ is the height level with 1 being the lowest level and $nlev$ being the highest level, $Z_z$ is the height at level $z$, and $CF_z$ is the cloud fraction at level $z$. As an example, if the 2D low level cloud fraction is 80%, and the lowest level clouds are located at 1000 m with a CF of 60%, and then the next height level at 1200 m has a CF of 80% that is equal to the low cloud fraction, the cloud base level would be (60/80)*1000 m + ((80 – 60)/80)*1200 m = 1050 m. However, a complication arises due to limited model vertical resolution. The existence of cloud at a point means that the cloud layer thickness is equal to the distance between the model

half levels bounding the point, but if the cloud was fully resolved, it could be anything greater than 0 up to this value. To account for this effect in comparisons to observations, cloud bases are computed separately using both the half levels directly below the cloud mass levels and using the cloud mass levels, and then these values are averaged as a best estimate. The same method is applied for estimating cloud top heights but integrating normalized CF-weighted height levels downward from above cloud top until the 2D low level cloud fraction is consumed. Cloud base and top temperatures are computed in the same way. The cloud

layer-mean $N_d$ is computed by averaging $N_d$ at each level in the cloud weighted by the CF at that level. Note that this is the grid-scale "stratiform" $N_d$ whereas LWP and CF include convective contributions to be consistent with observations. This cloud-layer mean $N_d$ is used in a dataset referred to as "E3SM" (see Table 1), whereas 2 additional model datasets are constructed to mimic TOA and surface observational retrievals, respectively.

The TOA-based $N_d$ retrieval used in the "E3SM TOA" dataset (Table 1) leverages simulated MODerate resolution Imaging

Spectroradiometer (MODIS) retrievals (Pincus et al. 2012) using the Cloud Feedback Model Intercomparison Project Observation Simulator Package (COSP; Bodas-Salcedo et al. 2011) version 2 (Swales et al. 2018). The simulator reads in E3SMv1 vertical profiles of layer COD and $R_{eff}$ for liquid and ice from which MODIS TOA visible and near infrared radiances are estimated. $R_{eff}$ is determined from the 2-moment parameterized cloud droplet size distribution in the Morrison-Gettelman (MG2) microphysics scheme (Morrison and Gettelman 2008, Gettelman and Morrison 2015, Gettelman et al. 2015). From these inputs and predicted

radiances, 2.1-µm $R_{eff}$ as estimated from a TOA perspective is predicted, from which the product of COD and $R_{eff}$ is be used to estimate LWP (Pincus et al. 2012). Simulated MODIS-retrieved LWP and COD with cloud top temperature are used to derive $N_d$ following equation 2 in section 2.1 that is also used for observational TOA retrievals. The simulated surface-based $N_d$ retrieval

used in the "E3SM Sfc" dataset (Table 1) uses vertically integrated cloud plus rain water content (since surface-based observations are also impacted by rain) with COSP-simulated COD and cloud depth following equation 1 in section 2.1 that is also used for surface-based observations. Cloud adiabaticity is also computed following the same method used for observations in section 2.2 by dividing LWP by the adiabatic LWP estimated from moist adiabatic ascent from cloud base to top.

To summarize the differences between the 3 E3SM datasets used in comparisons (E3SM, E3SM Sfc, and E3SM TOA) as highlighted in Table 1, E3SM Sfc uses direct model output like E3SM but with a $N_d$ retrieval following the surface-based $N_d$ retrieval for observations with directly predicted LWP and cloud depth coupled with COD obtained from COSP. E3SM TOA uses the same COSP COD as input to its $N_d$ retrieval but further uses LWP from the COSP MODIS simulator as input with a constant cloud adiabaticity of 0.8 following TOA-based observational retrievals. E3SM-predicted $R_{eff}$ and COD vertical profiles are inputs to the COSP MODIS simulator. However, LWP is predicted by the COSP MODIS simulator based on $R_{eff}$ and COD inputs, whereas LWP for E3SM and E3SM Sfc datasets is predicted and not linked by a simple adiabatic droplet growth model to $R_{eff}$ and COD. Thus, the relationship between COD, LWP, $R_{eff}$, and $N_d$ differs between datasets due to differing LWP inputs and $N_d$ retrieval assumptions, which motivates the usage of multiple retrievals with direct model output for better interpretable comparisons with observational retrievals.

## 2.4 Comparison Methods

All datasets and variables used in comparisons are listed in Table 1 including 5- and 60-minute averaged surface observations (Obs Sfc 5min, Obs Sfc 60min), 4-km and 1° satellite observations (Obs TOA 4km, Obs TOA 1deg) and E3SMv1 datasets that use direct output (E3SM), surface-estimated $N_d$ (E3SM Sfc) and TOA-estimated $N_d$ (E3SM TOA). For both observations and E3SM, the surface $N_d$ retrieval (equation 1) is the same as the TOA $N_d$ retrieval (equation 2) except that (i) no cloud adiabaticity assumption is made and (ii) the sources for LWP and COD inputs to equations 1 and 2 differ (Table 1). Effects of assuming constant adiabaticity on cloud susceptibilities are explored in analyses by making use of surface retrieval sensitivity tests. These tests still use surface-retrieved COD and LWP for observations and COSP COD and model-predicted LWP for E3SM when computing $N_d$ but use equation 2 rather than equation 1 with adiabaticity assumed to be constant at 80% to match TOA datasets. Although it would be ideal to apply a parameterization for adiabaticity within TOA retrievals to potentially improve their accuracy, deriving such a parameterization is beyond the scope of this study. The sensitivity tests instead are simply meant to quantify differences between constant and variable adiabaticity representations in retrievals. How cloud adiabaticity is handled in retrievals will be shown to be a key factor in understanding differences between the various model and observation datasets.

LWP estimates also slightly differ between the 3 E3SMv1 datasets. To be consistent with surface measurements, E3SM surface retrievals use total LWP inclusive of rain and convective liquid from the Zhang-McFarlane (Z-M) scheme (Zhang and McFarlane 1995) that accounts for 15% of the total LWP in single layer liquid cloud situations at ENA. The direct output E3SM dataset uses summed column-integrated grid scale and Z-M cloud water (not including rain water path), and TOA retrievals use COSP simulated LWP. Times with drizzle can influence the accuracy of retrievals (Cadeddu et al. 2020) but are not removed except for surface-based observations when drizzle at the surface is sufficient to obscure MWR LWP retrievals. For some variables such as albedo, CCN concentration, COD, and $R_{eff}$, the value from a single dataset is used for the others that do not have uniquely derived values (rightmost column in Table 1). For these variables, only differing sampling constraints produce slight differences in values between datasets. Since only overcast conditions are analyzed, all-sky albedo varies as cloud albedo varies, and thus we use all-sky values in analyses to avoid estimating clear sky contributions.

Comparisons between E3SMv1 and observational datasets are confined to specific situations to limit sources of retrieval uncertainty and possible contributors to dataset differences. All comparisons are limited to the column over the ENA site. In

E3SMv1 datasets, single layer liquid cloud situations are isolated by removing times with COSP-simulated MODIS IWP > 0 or cloud top temperature < 0°C. E3SMv1-predicted IWP is not used since it is commonly slightly greater than 0. This same IWP and cloud top temperature filtering is applied to VISST datasets, which allows for multiple liquid cloud layers to exist due to a lack of sufficient TOA data to remove such situations. However, surface-based vertically pointing radar measurements indicate that such situations are not common for the overcast, liquid cloud situations considered. For surface-based observations, only single layer cloud situations are considered, and situations with cloud top temperature < 0°C are removed using the radar-detected cloud boundaries described in section 2.2. Only situations with CF > 95%, solar zenith angle (SZA) < 65°, LWP > 20 g m$^{-2}$, and COD > 4 are included following some recommendations in Grosvenor et al. (2018) and Gryspeerdt et al. (2022). E3SMv1 and TOA measurements of these variables are spatial averages, whereas surface measurements of these variables are averaged over 5- and 60-minute periods. For analyses relating cloud properties to surface CCN concentrations, only times with cloud base potential temperature < 2°C warmer than the near surface potential temperature are included. These are situations that are most likely to have surface-coupled cloud bases that respond to surface CCN more strongly than decoupled clouds (Dong et al. 2015). This threshold will not remove all uncoupled clouds, but it allows for retaining of greater sample sizes. Other cloud-surface coupling indices produced similar results as the potential temperature metric (Fig. S1). Lastly, Graciosa Island where the ENA site is located is not represented in the E3SMv1 simulation but has been shown to influence the boundary layer vertical motion and turbulence at the site when wind directions are between 90° and 310° where 0° is from the north (Ghate et al. 2021, Jeong et al. 2022). It is also possible for the terrain on the island reaching 400 m ASL to influence clouds. Such potential island effects have not been removed from analyses to retain sufficient sampling but could contribute to some of the differences between observations and E3SMv1.

**3 Cloud, Aerosol, Radiation, and Atmospheric State Properties**

The sampling of overcast single layer liquid clouds at ENA depends on the resolution and sensitivity of each dataset. The average warm, liquid CF and percentage of times with CF > 95% for times without supercooled and ice clouds are greater for the 5-minute surface and 4-km TOA retrievals than the 60-minute surface and 1° TOA retrievals (Table 2). This is likely the result of increasing probability of encountering overlying cloud layers as scale increases. The average warm, liquid CF in E3SMv1 for times without overlying clouds is slightly lower than observed. However, the percentage of time that CF exceeds 95% is similar for satellite-observed and COSP-simulated TOA estimates (20 vs. 23%) with the 60-min surface estimate a bit lower (17%) and direct model output having a greater occurrence (31%). SZA, LWP, and COD constraints further reduce observational hourly sampling to between 649 and 1,381 with between 1,697 and 1,941 hourly E3SM samples depending on the dataset. Analyses involving surface CCN concentration have even fewer samples. This is partly because of the surface coupling constraint, though observed samples drop further than for E3SM due to some missing and bad CCN data, resulting in 197-328 hourly observational samples but 1,303-1,459 E3SM samples.

For times with overcast, warm liquid clouds and sufficient SZA, distributions of TOA albedo, cloud effective albedo at the surface, and COD are shown in Fig. 2. E3SMv1 has slightly greater median TOA albedos than observed (Fig. 2a and 3), though median surface-estimated cloud effective albedos are slightly lower (Fig. 2b and 3). This TOA difference will be shown later to be the result of differing SZA-$N_d$ correlations between E3SMv1 and observations. Indeed, median COD is similar between E3SMv1 and TOA observations (Fig. 3). Surface-estimated COD values are greater than simulated, consistent with greater cloud effective albedos caused by greater LWPs being sampled with similar $N_d$ values (Fig. 3 and 4a-b). Median LWP is lesser in E3SMv1 as compared to observations (Fig. 3 and 4a) by ~30% (62-68 vs. 78-92 g m$^{-2}$). Unlike LWP, $N_d$ is notably greater in E3SMv1 TOA than TOA observations (95 vs. 56-61 cm$^{-3}$ medians; Fig. 3and 4b) and droplet $R_{eff}$ is notably smaller (9.6 vs. 13 μm median values;

Fig. 3 and 4c) though E3SMv1 direct outputted $N_d$ is lesser with a median value of 70 cm$^{-3}$. While surface-retrieved $R_{eff}$ observations are also greater than simulated, 60-min surface-retrieved $N_d$ observations are similar to E3SMv1 surface-retrieved $N_d$, both having median values near 110 cm$^{-3}$ (Fig. 3 and 4b-c), while 0.2% surface CCN concentration distributions are also similar (Fig. 3 and 4d). Model-observation $N_d$ comparisons clearly depend tremendously on how $N_d$ is retrieved with typical values changing by up to a factor of 2 based on retrieval method. This emphasizes the importance of using multiple different retrievals to assess and better interpret model-observation differences.

For overcast cloud conditions at ENA, E3SMv1 has weaker than observed inversion strengths and higher than observed above inversion relative humidity (Fig. 5a-b). Even though the simulation winds are nudged to MERRA2, thermodynamic state is not and can have errors develop. These errors suggest excessive mixing between the boundary layer and free troposphere or insufficient free tropospheric subsidence. Zheng et al. (2020) found insufficient vertical mixing in E3SMv1 for the subtropical stratocumulus to cumulus transition region of the northeast Pacific, but Ma et al. (2022) found excessive turbulent mixing in E3SMv1 that has been tuned down in E3SMv2 (Golaz et al. 2022). These inversion differences are associated with clouds in E3SMv1 that have similar cloud bases to observed (Fig. S2a-b) but higher and colder cloud tops (Fig. S2c-d) with greater cloud depths (Fig. 5c). TOA estimates of cloud depth are shallower than in surface retrievals and E3SM direct output due to assuming that clouds are 80% adiabatic whereas most clouds have lesser cloud adiabaticity, particularly in E3SM (Fig. 5d). Note that observationally estimated adiabaticities can exceed 100% and even approach 200% (Fig. 1). This typically occurs for thin clouds with low LWP values for which the LWP and cloud depth retrieval errors can cause adiabaticity errors on the order of 100%. However, most of the sampled observed clouds are subadiabatic, consistent with the results of Wu et al. (2020b), with a mean of 71% that is only slightly higher than the 63% found in Merk et al. (2016) and not far below the 80% assumed in TOA retrievals. However, E3SMv1's mean adiabaticity is 27%, a much lesser value than observed that is associated with its deeper clouds, weaker inversions, but similar LWPs. 68% of simulated clouds have an adiabaticity < 30%, whereas only 16% do in observations (Fig. 5d). On the other hand, 64% of observed clouds are more than 60% adiabatic but only 12% of clouds sampled in E3SMv1 reach this threshold. Thus, an assumption of 80% adiabaticity for E3SM clouds is substantially biased high. The differences in how cloud adiabaticity is handled across the datasets will be shown to impact quantification of susceptibilities.

## 4 Cloud Albedo Susceptibility

Differences in observed and simulated cloud properties can result from differences in atmospheric states and/or errors from subgrid scale parameterizations. Such errors do not necessarily imply that the responses of clouds and radiation to aerosol perturbations are incorrect, and it is these responses that are the primary focus here. In particular, the response of cloud albedo to CCN is evaluated in this section. The cloud albedo ($A$) susceptibility to changes in CCN number concentration is evaluated by separating the Twomey effect and LWP response components with Equation 4 (Quaas et al 2008):

$$\frac{dA}{dlnCCN} = \left( \frac{\partial A}{\partial lnN_d} + \frac{\partial A}{\partial lnLWP} \frac{dlnLWP}{dlnN_d} \right) \frac{dlnN_d}{dlnCCN}, \tag{4}$$

Given the overcast cloud condition requirement, the CF response is neglected, and we analyze all sky albedo for TOA relationships. COD susceptibility is also analyzed, for which $A$ in equation 4 is simply replaced with lnCOD. Changes in the SZA affect $A$ via changes in the slant path of shortwave radiation through the cloud, whereas COD estimates the vertical component of the change in shortwave radiation.

The following sections quantify the individual terms of Equation 4 within each of the E3SMv1 and observation datasets within the context of retrieval uncertainties. Analyses are also performed to assess possible causes for model-observation discrepancies

including differences in $R_{eff}$ and cloud adiabaticity. Within the context of equation 4, $R_{eff}$ is implicit since it is retrieved from only 2 variables (LWP and COD) for surface measurements, while for TOA measurements, COD and $R_{eff}$ collectively determine LWP. These variables together also determine cloud layer mean $N_d$ with an assumption of scaled adiabatic growth of droplets from cloud base to top. In E3SM, $R_{eff}$ also only depends on 2 variables, the predicted LWC and $N_d$ that dictate the 2-moment cloud droplet size distribution. Thus, LWP and $N_d$ encapsulate the information content available in the cloud retrievals. Surface retrievals of cloud depth provide additional information on the cloud adiabaticity, which scales adiabatic $N_d$ and LWP by $\sqrt{\alpha}N_{d,adiabatic}$ and $\alpha LWP_{adiabatic}$, producing equation 5:

$$\frac{dA}{dlnCCN} = \left( \frac{\partial A}{\partial \ln(\sqrt{\alpha}N_{d,adiabatic})} + \frac{\partial A}{\partial \ln(\alpha LWP_{adiabatic})} \frac{dln(\alpha LWP_{adiabatic})}{dln(\sqrt{\alpha}N_{d,adiabatic})} \right) \frac{dln(\sqrt{\alpha}N_{d,adiabatic})}{dlnCCN}, \tag{5}$$

Hence, the magnitudes of terms can change based on the chosen constant $\alpha$ value, and because $0 < \alpha \leq 1$, LWP decreases faster than $N_d$ from adiabatic values as $\alpha$ decreases. However, we will show that retrieved $\alpha$ is not constant and varies with $N_d$ and LWP, which alters the magnitudes of terms further and will be shown to be important in understanding differences between datasets.

### 4.1 Twomey Effect Comparison

We first evaluate the Twomey effect signified by $\frac{\partial A}{\partial lnN_d} \frac{dlnN_d}{dlnCCN}$, which describes the response of albedo to a change in $N_d$ via a change in CCN.

### 4.1.1 Albedo Dependence on Droplet Concentration

Isolating the Twomey effect requires accounting for the effect of changes in LWP on $A$. The effect of both LWP and $N_d$ on $A$ is visualized in Fig. 6a-d and 7a-d, which show heatmaps of median $A$ within bins of $N_d$ and LWP for the various datasets analyzed. All datasets show similar patterns with $A$ increasing foremost with increasing LWP and secondarily with increasing $N_d$, though the $A$ sensitivity to $N_d$ is muted in E3SMv1, particularly for TOA estimates. Indeed, absolute differences between E3SMv1 and observations in Fig. 6e-f and 7e-f show that E3SMv1 $A$ is much greater than observed for relatively low $N_d$ values, a difference that decreases as $N_d$ increases. $\frac{\partial A}{\partial lnN_d}$ is further quantified by multiple linear regression. Regression coefficients confirm that $\frac{\partial A}{\partial lnLWP}$ is greater than $\frac{\partial A}{\partial lnN_d}$ by a factor of 2-8 depending on the dataset considered (Fig. 8). All dataset fits have Pearson correlation coefficients (r) of 0.81-0.88 (Fig. 8), showing that LWP and $N_d$ alone predict much of the $A$ variability. Consistent with Fig. 6-7, E3SMv1 coefficients have a similar response of $A$ to LWP as observations but the response of $A$ to $N_d$ is about half that observed. The estimated response of $A$ to $N_d$ also depends on how $N_d$ is retrieved. Whereas a surface-retrieved $N_d$ value decreases $\frac{\partial A}{\partial lnN_d}$ relative to the actual E3SMv1-predicted $N_d$ value (green vs. blue open diamonds), a TOA-retrieved $N_d$ does the opposite (orange vs. blue open circles) (Fig. 8). This difference is explained by differences in cloud adiabaticity assumptions made by TOA and surface retrievals. TOA retrievals assume an adiabaticity of 80% while surface retrievals allow adiabaticity to vary by leveraging cloud depth measurements. If $N_d$ is recomputed from surface retrieved COD and LWP assuming an adiabaticity of 80% in equation 2, then the sensitivity of albedo to $N_d$ increases (light green relative to dark green symbols in Fig. 8). The increase is especially dramatic for E3SMv1, where surface-retrieved $\frac{\partial A}{\partial lnN_d}$ assuming 80% adiabaticity now exceeds the value derived from direct model output in agreement with the higher TOA retrieval value relative to direct model output. Clearly cloud adiabaticity can affect $\frac{\partial A}{\partial lnN_d}$, a topic discussed further in the next section. While the resolution of the $N_d$ retrieval input data does not affect TOA $\frac{\partial A}{\partial lnN_d}$, the surface $\frac{\partial A}{\partial lnN_d}$ decreases by 15% when switching from 5-min to 60-min inputs, a difference that is less than the model-observation

differences (Fig. 8). Thus, there is agreement between datasets that E3SMv1 significantly underestimates the $N_d$ effect on $A$ despite reasonable sensitivity of $A$ to LWP.

### 4.1.2 Factors Affecting Model-Observation Differences

We first investigate how SZA impacts differences in observed and simulated $\frac{\partial A}{\partial lnN_d}$. SZA and $N_d$ are not correlated in observations,

but E3SMv1 SZA decreases as $N_d$ increases (Fig. S3). This difference between E3SMv1 and observations is evident in Fig. 9 where E3SMv1 has greater SZA values for relatively low $N_d$ and lesser SZA values for relatively high $N_d$. We first check whether this correlation is caused by the diurnal cycle. There is an early afternoon $N_d$ minimum in E3SMv1 datasets (dashed lines in Fig. S4e) that correlates with a SZA minimum (Fig. S4b) whereas observed variables lack such a correlation. However, this is the opposite SZA-$N_d$ correlation in Fig. S3 and 9, so this cannot be the cause of the SZA-$N_d$ correlation in those figures. Surface CCN

concentration reaches a minimum in early afternoon (Fig. S5a), which is less apparent in observations than E3SMv1 and could be the cause for the simulated early afternoon $N_d$ minimum. We next investigate the seasonal cycle. Simulated $N_d$ strongly peaks in July with median values that are more than twice the wintertime minimum (Fig. 10e), a seasonal cycle that is anti-correlated with the SZA and $A$ seasonal cycles (Fig. 10b-c). $R_{eff}$ also exhibits a notable seasonal cycle, but LWP does not. Thus, seasonality is likely the cause for the E3SMv1 SZA-$N_d$ correlation. Observations also exhibit $N_d$ seasonality consistent with past studies (e.g.,

Wood et al. 2015) but with a peak in May that does not correlate with the SZA seasonal cycle. Both E3SMv1 and observations also exhibit strong seasonal cycles in surface CCN concentration (Fig. S5b), and despite being constrained to specific cloud situations, the wintertime minimum and summer maximum in CCN concentration are consistent with the results of Zheng et al. (2018). The observed maximum is 2 months earlier than simulated, possibly contributing to the ~2-month earlier peak in observed maximum $N_d$ relative to E3SMv1 that leads to the differing relationship of $N_d$ with SZA between E3SMv1 and observations.

Because seasonality influenced SZA-$N_d$ correlations significantly impact model-observation differences in cloud albedo susceptibility terms, we also analyze the COD response to $N_d$ and LWP. COD is much less influenced by such correlations, which allows for better isolation of cloud radiative effects caused by $N_d$ and LWP alone rather than SZA. Observed and simulated $\frac{\partial lnCOD}{\partial lnLWP}$ values are similar, which is consistent with $\frac{\partial A}{\partial lnLWP}$ comparisons (Fig. 11). Whereas observed and simulated $\frac{\partial A}{\partial lnN_d}$ values significantly differed, observed and simulated $\frac{\partial COD}{\partial lnN_d}$ values are much more similar (Fig. 11). The difference that remains is E3SM

and E3SM Sfc values being less than surface-based observations. However, controlling for cloud adiabaticity differences between E3SM and observations removes this difference (light green symbols in Fig. 11). As described in section 4.1.1, this is done by recomputing $N_d$ in surface retrievals with scaled cloud depths to values that would be associated with 80% adiabatic LWP. This increases $\frac{\partial lnCOD}{\partial lnLWP}$ by over 10% for observations and almost 50% for E3SMv1, leading to values that are almost identical to TOA retrieved values that assume 80% adiabaticity. To visualize this, Fig. 12 shows absolute differences of lnCOD between E3SM and

observational datasets as a function of LWP and $N_d$ (lnCOD values for each dataset are shown in Fig. S6). Note that when the exact same retrieval with a constant adiabaticity is applied to both E3SM and observations (Fig. 12a, c), there is virtually no difference in lnCOD for a given LWP and $N_d$ because $N_d$ is computed from COD, LWP, and adiabaticity or cloud depth. When adiabaticity is not held constant, lnCOD values for surface-based retrievals in E3SMv1 and observations diverge (Fig. 12b). Why is this? As shown in Fig. 13, cloud adiabaticity is frequently lower than 80% and decreases as $N_d$ decreases and LWP increases.

E3SMv1 also has much more subadiabatic clouds than observed. When $N_d$ is recomputed using a constant adiabaticity, it causes a shift in the $N_d$ distribution. For the case of constant 80% adiabaticity, low $N_d$ values with adiabaticity much less than 80% increase more than higher $N_d$ values that have higher adiabaticity values. This causes a narrowing of the $lnN_d$ distribution, which increases

$\frac{\partial lnCOD}{\partial lnN_d}$ (Fig. 14). Because adiabaticity also decreases with increasing LWP, the shift in the $lnN_d$ distribution varies by LWP (Fig.

14), causing a slight decrease in $\frac{\partial lnCOD}{\partial lnLWP}$ (Fig. 11) that can also affect $\frac{\partial lnCOD}{\partial lnN_d}$. Thus, the grossly subadiabatic clouds in E3SMv1

suppress the change of COD with $N_d$ relative to more adiabatic observed clouds and retrievals with constant 80% adiabaticity.

    Though simulated and observed values of $\frac{\partial lnCOD}{\partial lnN_d}$ agree once cloud adiabaticity effects on $N_d$ retrievals are removed, lnCOD

is always greater than observed for the E3SM direct model output (Fig. 12d-e). This is potentially the result of lesser $R_{eff}$ values in

E3SMv1 than observed for given LWP and $N_d$ values (Fig. 15c-d; $R_{eff}$ values for each dataset are shown in Fig. S7). While $R_{eff}$ is

also systematically lower for given LWP and $N_d$ values in TOA retrievals (Fig. 15a), this does not lead to COD differences as a

function of LWP and $N_d$ in the TOA datasets because the exact same $N_d$ calculation is used for Obs TOA and E3SM TOA with

sole dependence on LWP, COD, and the constant $k$ parameter that relates droplet volume mean radius to $R_{eff}$. The same applies for

surface retrievals (Fig. 15b), but non-systematic differences are created by differing adiabaticity values and surface retrieved $R_{eff}$

that is sensitive to the entire cloud layer whereas E3SM Sfc $R_{eff}$ reflects near cloud top values obtained from COSP. That E3SM's

$R_{eff}$ is lower in direct model output than retrievals means that the parameterized size distribution differs from that assumed in

observations. Although it is possible that remote sensing retrieved $R_{eff}$ is high biased, recent studies show limited satellite retrieval

biases when compared with in situ measurements (Witte et al. 2018, Kang et al. 2021). In addition, aircraft in situ measurements

from the ACE-ENA campaign (Wang et al. 2022) support remotely sensed $R_{eff}$ values being greater in observations than E3SMv1

with corresponding lower $N_d$ values (Wu et al. 2020a). These results provide further support that the simulated SZA-$N_d$ correlation

caused by seasonality plus simulated cloud adiabaticity that differs from observations mute the effect of $N_d$ on $A$ in E3SMv1, while

smaller $R_{eff}$ in E3SMv1 than observed amplify $A$ for a given $N_d$.

### 4.1.3 Aerosol Activation into Cloud Droplets

Twomey effect differences between E3SMv1 and observations also depend on the response of $N_d$ to CCN ($\frac{dlnN_d}{dlnCCN}$). This term is

expected to strongly depend on aerosol activation, though with modulation by $N_d$ sinks such as evaporation and precipitation. It is

only evaluated for situations in which clouds are more likely to be coupled with the surface where CCN measurements are made.

Although Jones et al. (2011) use a threshold of 0.5°C for the difference in cloud base and near surface potential temperature, we

increase this to 2°C to retain more samples and to account for uncertainty in potential temperature measurements obtained from

interpolated soundings that are often separated by 12 hours.

    Figure 16 shows $N_d$ as a function of surface 0.2% CCN for all datasets, and correlation coefficients from Theil-Sen robust

linear regression are provided in Fig. 17. The sensitivity of $N_d$ to CCN is greater for E3SM direct output than any other dataset.

The sensitivity reduces for E3SM TOA but remains greater than Obs TOA datasets, and it reduces further for E3SM Sfc, which

aligns well with Obs Sfc datasets. The fit is weaker in observations than E3SMv1 as shown by the spread of $N_d$ values for a given

CCN value (Fig. 16) and the correlation coefficients in Fig. 17 (0.29-0.54 for observations vs. 0.39-0.85 for E3SMv1 datasets). A

substantial portion of this spread is a result of errors in TOA- and surface-retrieved $N_d$ values. This is shown by the greater spread

in the COSP- and surface-simulated E3SMv1 $N_d$ relationships as compared to the direct $N_d$ relationship, quantified by differences

in E3SMv1 correlation coefficients (0.85 (direct) vs. 0.65 (satellite) and 0.39 (surface)) as compared to observational coefficients

(0.46-0.54 (TOA) and 0.29-0.30 (surface)). Why do surface retrieved $\frac{dlnN_d}{dlnCCN}$ values agree between E3SMv1 and observations while

TOA values do not? As for $\frac{\partial lnCOD}{\partial lnN_d}$ discrepancies, this contrast is caused by variable cloud adiabaticity in surface retrievals and

constant 80% adiabaticity in TOA retrievals. When surface retrieved $N_d$ is recomputed assuming 80% adiabaticity, E3SMv1's $N_d$

sensitivity to CCN substantially increases, but the observed sensitivity does not (dark green to light green symbols in Fig. 17).

Thus, once the effect of cloud adiabaticity differences between E3SMv1 and observations on $N_d$ are removed, it becomes clear that the sensitivity of $N_d$ to CCN is too high in E3SMv1. Aircraft observed $N_d$ vs. CCN concentration during ACE-ENA also supports a conclusion of greater sensitivity in E3SMv1 (Tang et al. 2023). This explains why surface-retrieved $N_d$ values agree between E3SM Sfc and Obs Sfc but are greater for E3SM TOA than Obs TOA. Lower adiabaticities in E3SMv1 than observed lower $N_d$ further from their adiabatic value, but when that effect is removed, it becomes clear that E3SMv1 has higher $N_d$ values than

observed despite reasonable CCN values because of a greater sensitivity of $N_d$ to CCN.

     These comparisons suggest that aerosol activation in E3SMv1 is potentially too high. This is consistent with the findings of Ghan et al. (2011) for the Abdul-Razzak Ghan (ARG) scheme (Abdul-Razzak and Ghan 2000) used in E3SM. Gong et al. (2023) also find that the ARG scheme coupled with the Cloud Layers Unified by Binormals (Golaz et al. 2002, Larson and Golaz 2005) and the four-mode Modal Aerosol Module (Liu et al. 2016) in the Community Earth System Model version 2.1 with the Community

Atmosphere Model version 6 (Danabasoglu et al. 2020) produces greater cloud supersaturations than retrieved from observations at ENA. With a similar setup in E3SMv1, this could also be influencing the $\frac{dlnN_d}{dlnCCN}$ differences shown here. However, it is also possible that unrealistic $N_d$ sinks including precipitation and evaporation contribute to $\frac{dlnN_d}{dlnCCN}$ being greater in E3SMv1 than observed, and more investigation into these processes is required.

     E3SMv1-observation $\frac{dlnN_d}{dlnCCN}$ differences offset $\frac{\partial A}{\partial lnN_d}$ differences for surface retrievals, producing Twomey effect ($\frac{\partial A}{\partial lnN_d}\frac{dlnN_d}{dlnCCN}$)

and albedo susceptibility estimates that are similar (Fig. 18). For TOA estimates, $\frac{\partial A}{\partial lnN_d}$ differences are too great to overcome, leading to a weaker than observed Twomey effect and albedo susceptibility best estimate that is negative (Fig. 18). As discussed in section 4.1.2, this is caused by the negative SZA-$N_d$ correlation in E3SM that is not observed and suppresses $\frac{\partial A}{\partial lnN_d}$ relative to observations. Removing SZA-$N_d$ correlation effects by instead examining COD susceptibility shows in fact that the Twomey effect ($\frac{lnCOD}{\partial lnN_d}\frac{dlnN_d}{dlnCCN}$) is greater in E3SMv1 than observed, though this is only true for surface retrievals once model-observation cloud

adiabaticity differences are removed. Removing adiabaticity effects gives Twomey effects that are about 30-40% greater in E3SMv1 than observed, while lnCOD susceptibility can be up to a factor of 2 greater depending on dataset (Fig. 19).

**4.2 LWP Susceptibility Comparison**

Since only overcast cloud conditions are considered, the second aerosol indirect effect is confined to the LWP susceptibility to $N_d$ ($\frac{dlnLWP}{dlnN_d}$), which can accentuate or mute the Twomey effect. Figure 20 shows $N_d$-bin normalized joint distributions $N_d$ and LWP

for the various datasets analyzed. Like the Twomey effect, the quantified LWP responses depend on the datasets considered since LWP and $N_d$ values shift depending on how they are retrieved. Somewhat poor linear fits are readily apparent, as highlighted by low correlation coefficients (0.21-0.58) in Fig. 21. E3SM susceptibilities range from -0.3 to -0.4, which is similar to those in surface observations, while TOA observations have weaker susceptibilities around -0.2. Removing cloud adiabaticity effects in surface retrievals results in a weaker LWP susceptibility in E3SM than observations, while the opposite is true for TOA retrievals. Thus,

there is no consensus among comparisons, and all that can be said is that E3SMv1 has similar values to observed, which differs from most GCMs that produce a positive LWP susceptibility (Quaas et al. 2009, Gryspeerdt et al. 2020).

     Previous satellite retrieval and LES studies discussed in the introduction have highlighted an inverted V response of LWP to $N_d$ which has been hypothesized to be caused by drizzle suppression at relatively low $N_d$ values increasing LWP but entrainment-driven evaporation mechanisms reducing LWP for relatively high $N_d$ values in which non-drizzling clouds. To assess whether

LWP responses to $N_d$ change as $N_d$ increases, linear fits are applied for 2 different ranges of $N_d$ shown as dashed black lines in Fig.

20a-g and quantified in Fig. 21. Observational datasets become more negative as $N_d$ increases in agreement with past satellite retrieval studies. However, the opposite trend exists in E3SMv1 direct output and TOA datasets where LWP responses become less negative as $N_d$ increases, with the caveat that uncertainty is high for $N_d < 50$ cm$^{-3}$ due to limited sampling that is possibly associated with constraining analyses to overcast conditions. For surface retrievals, E3SM's LWP susceptibility becomes more negative with increasing $N_d$ like observed, but once cloud adiabaticity is held constant, the opposite occurs in disagreement with observations. Thus, the change in E3SM's LWP susceptibility with increasing $N_d$ is generally opposite to the expectation from proposed physical mechanisms in past observational and LES studies, indicating possibly different mechanisms that cause its overall negative sign than in observations.

### 4.2.1 Potential Physical Mechanisms Affecting Model-Observation Comparisons

Are negative LWP susceptibilities caused by physical processes, and if so, are they the same in observations and E3SMv1? If clouds are responding to the entrainment-evaporation mechanism, one might expect the LWP susceptibility to change with the inversion strength and above inversion RH. To assess this, the median LWP susceptibility is plotted in joint EIS-700-hPa RH bins in Fig. 22. No clear dependencies on EIS or RH arise in surface observations (Fig. 22a), while the LWP response becomes less negative as RH increases in TOA observations (Fig. 22c), consistent with more widespread satellite measurements of marine warm clouds in Chen et al. (2014). However, the simulated LWP response to $N_d$ becomes more negative as RH increases for a given EIS (Fig. 22b, d, e), which indicates that entrainment-driven evaporation is likely not a major control on the negative LWP response in E3SMv1.

However, it is apparent from the sample size contours in Fig. 22 that E3SM has lesser occurrences of strong inversions (high EIS) with low 700-hPa RH conditions than observed, consistent with Fig. 5. For a given LWP and $N_d$ value, E3SMv1 can have substantially deeper clouds than observed, often by a couple hundred meters (Fig. S8), and weaker inversions in E3SMv1 are present for all LWP-$N_d$ combinations (Figs. S9-10). While E3SM direct model output and surface retrievals have inversions that strengthen as $N_d$ increases with clouds that become shallower, the EIS gradient is notably absent in TOA datasets and the cloud depth dependence on $N_d$ changes sign. This is likely associated with the lack of variable cloud adiabaticity in TOA datasets and serves as caution for using such datasets alone to infer physical processes and evaluate model output. Past studies have shown that weaker stability with boundary layer and cloud deepening causes the LWP susceptibility to become more negative (e.g., Possner et al. 2020, Zhang et al. 2022). Thus, one might expect a more negative LWP susceptibility in E3SMv1 than observed if it were properly representing processes controlling the LWP response to $N_d$, but this is not the case, again suggesting that there are other causes for the negative simulated sign.

Another factor affecting LWP susceptibility is cloud adiabaticity (dark vs. light green symbols in Fig. 21). In observations, clouds become more adiabatic as $N_d$ increases, potentially due to drizzle suppression (Fig. 13). However, clouds only become slightly more adiabatic as $N_d$ increases in E3SMv1 (Fig. 13). Median hourly surface rain rates as a function of LWP and $N_d$ are shown in Fig. 23 with rain rates less than 0.001 mm h$^{-1}$ removed due to limited observational sensitivity. Observed clouds with less than 100 g m$^{-2}$ LWP or greater than 50 cm$^{-3}$ $N_d$ usually do not have rain reaching the surface, but this is common in E3SMv1 where sensitivity of surface rain rate to $N_d$ and LWP are muted. The hourly surface rain rate exceeds 0.001 mm h$^{-1}$ 62% of the time in E3SMv1 as compared to 15-18% in disdrometer and optical rain gauge observations, while rates exceed 0.01 mm h$^{-1}$ nearly half the time in E3SMv1 as compared to ~10% in observations (Table S1). Total surface rainfall accumulations are also nearly 3 times greater in E3SMv1 as evidenced by average rain rates inclusive of all times in Table S1. Too frequent and light precipitation is a well-known problem in GCMs (Stephens et al. 2010, Song et al. 2018) including E3SM (Ma et al. 2022) in which Zheng et al. (2020) has noted biases associated with the long microphysics time step and precipitation fraction parameterization. For the single

layer, overcast liquid clouds at ENA assessed here, 84% of the surface rainfall is produced in the Z-M convection parameterization (Table S1) while the frequency of stratiform drizzle is closer to observed. Figure S11 shows that surface hourly rain rates compare much more favorably when only stratiform rain rates for E3SM and E3SM Sfc are considered, indicating the Z-M parameterization is a primary driver of the drizzle frequency and amount biases. This is partly caused by deeper clouds in E3SM that more heavily precipitate (Fig. S12), likely a result of weaker inversions in E3SM. However, even for given cloud depth and $N_d$ values, clouds precipitate more heavily (Fig. S12) and are less adiabatic (Fig. S13) than observed. It is probable that the excessive convective drizzle in E3SMv1 contributes to much less adiabatic clouds than observed, a difference that increases as $N_d$ increases when observed drizzle ceases but E3SMv1 clouds continue to drizzle with only slightly lesser rates (Figs. 13, 23, and S12-13). It's possible that this muted sensitivity of rain rate and adiabaticity to $N_d$ contributes to the negative relationship of LWP with $N_d$. Cadeddu et al. (2023) also find that entrainment could be a primary controlling factor of cloud adiabaticity at ENA. How much entrainment affects simulated adiabaticity is not clear, but it is possible that dominant mechanisms controlling adiabaticity and its effects on LWP and $N_d$ differ between E3SMv1 and the real world. More research is needed to better understand how rain rate, entrainment, LWP, and $N_d$ interact to affect albedo susceptibility in both models and the real world.

A final important consideration for interpreting LWP susceptibility quantification is that the LWP-$N_d$ joint distribution is likely not entirely due to $N_d$ effects on LWP. Confounding factors such as meteorology or mesoscale cloud structures may combine with microphysical processes to create overall negative LWP-$N_d$ relationships. Indeed, present day minus pre-industrial E3SMv1 simulations produce a positive rather than negative LWP susceptibility (Christensen et al. 2023). Albedo and COD susceptibilities are also generally slightly less than Twomey effects ( $\frac{\partial A}{\partial lnN_d}\frac{dlnN_d}{dlnCCN}$ , $\frac{\partial lnCOD}{\partial lnN_d}\frac{dlnN_d}{dlnCCN}$ ) despite LWP response ( $\frac{\partial A}{\partial lnLWP}\frac{dlnLWP}{dlnN_d}\frac{dlnN_d}{dlnCCN}$ , $\frac{\partial lnCOD}{\partial lnLWP}\frac{dlnLWP}{dlnN_d}\frac{dlnN_d}{dlnCCN}$ ) estimates that are large enough to completely cancel the Twomey effects (Figs. 18-19). This suggests that spread in the $\frac{dlnLWP}{dlnN_d}$ and $\frac{dlnN_d}{dlnCCN}$ relationships and unaccounted for covariations between lnLWP and lnCCN partly counteract the estimated LWP response. Thus, further research is needed to develop methods for sufficiently evaluating cloud adjustments in GCMs using present day observations.

**5 Conclusions**

Surface-based and satellite observations collected at the ARM ENA site in the Azores over 5 years were used to evaluate factors controlling single layer liquid cloud albedo susceptibility for overcast cloud conditions in E3SMv1. While a single geographical location such as the ENA site is not representative of all environments and cloud types, its relatively comprehensive measurements and retrievals including CCN number concentration, atmospheric stability and humidity, radiative fluxes, and cloud LWP, $N_d$, rain rate, and cloud adiabaticity in a marine environment with frequent single layer liquid clouds susceptible to aerosol influences allows for detailed model evaluation targeting specific processes. This methodology provides valuable context for more traditional model evaluation via longer-term, global metrics. Surface retrievals with 5- and 60-min resolutions, geostationary satellite retrievals with 4-km and 1° resolutions, and 1° resolution E3SMv1 datasets using different $N_d$ and LWP retrievals from raw model output, COSP satellite simulator output, and surface variables, are all analyzed to assess the robustness of model-observation comparisons.

Simulated cloud albedo values are greater for given LWP and $N_d$ values due to $R_{eff}$ being smaller than retrieved in observations. However, cloud albedo sensitivities to $N_d$ and LWP are well simulated by E3SMv1 after accounting for a simulated SZA-$N_d$ correlation caused by seasonality that is not observed by analyzing COD susceptibility and after accounting for differences in observed and simulated cloud adiabaticity. Both effects mute the simulated sensitivity of cloud albedo to $N_d$. E3SMv1 adiabaticities

are much lower than observed and the common 80% assumed in satellite retrievals, and adiabaticities affect the sensitivity of cloud albedo to $lnN_d$ by altering the width of the $lnN_d$ distribution. LWP is also less than observed while $N_d$ is greater than observed if cloud adiabaticity differences between E3SMv1 and observations are removed, despite a similar overall distribution of surface

CCN concentration to observed. Because of logarithmic sensitivities, this means that linear perturbations in $N_d$ will result in a lesser Twomey effect in E3SMv1 compared to observations at ENA, though perturbations in LWP could potentially result in a greater effect.

Greater $N_d$ values in E3SMv1 may result from excessive activation of CCN. This drives a Twomey effect that is too large by 30-40% when using TOA and surface retrieved $N_d$ values. However, this difference again only emerges after controlling for SZA

and cloud adiabaticity. LWP decreases as $N_d$ increases like observed, something that is not produced by most GCMs. However, the LWP susceptibility becomes more negative in E3SMv1 as above inversion RH increases, a trend that is not observed and not consistent with drier conditions facilitating more evaporation. In addition, the simulated LWP susceptibility does not become more negative as $N_d$ increases like observed. Thus, it is unlikely that an entrainment-driven evaporation mechanism is operating in E3SMv1. E3SMv1 also has weaker inversions and deeper clouds than observed, which partly contributes to excessive convectively

driven drizzle that is a common bias in GCMs (Stephens et al. 2010), though a substantial bias remains after controlling for cloud depth. There is also little decrease in drizzle rate and little increase in cloud adiabaticity as $N_d$ increases for a given LWP or cloud depth in E3SMv1, which is in stark contrast to observations. Because of these major differences in observed and simulated cloud properties, the causes for the similar negative LWP susceptibility in E3SMv1 and observations likely differ. The negative LWP responses also only slightly decrease the overall cloud albedo susceptibilities, especially for E3SMv1, indicating that there are

confounding effects that render a statistical correlation between LWP and $N_d$ insufficient for assessing $N_d$ impacts on LWP.

Because E3SMv1 has a greater Twomey effect than observed and because adjustments lower the COD susceptibility more in observations than E3SMv1, simulated COD susceptibility values are about double those observed after controlling for cloud adiabaticity differences, which is consistent with overly negative ERFaci assessed in previous studies (Golaz et al. 2019, Rasch et al. 2019, Wang et al. 2020, Ma et al. 2022, Zhang et al. 2022). This also indicates that greater $\frac{dlnN_d}{dlnCCN}$ in E3SMv1, as modulated by

aerosol activation, $N_d$ evaporation, and precipitation, along with lesser muting by adjustments, are the primary drivers of greater radiative susceptibilities to aerosols in E3SMv1.

Differences between E3SMv1 and observations often exceed the observational spreads after critically controlling for cloud adiabaticity model-observation differences, but the spreads are still substantial. Both retrieval scale and simplifying assumptions contribute similar magnitudes of uncertainty in quantifying cloud albedo susceptibility and the Twomey effect, while retrieval

assumptions contribute the most uncertainty in quantifying the LWP susceptibility. A major contributor to these discrepancies is the retrieval of $N_d$. For the most accurate $N_d$ retrieval, it is best to compute $N_d$ at the highest resolution possible and average it to coarser scales (McComiskey and Feingold 2012, Feingold et al. 2022). However, because E3SM surface and TOA $N_d$ retrievals were performed using E3SMv1 1° output, 1° and 60-min observational $N_d$ retrievals were instead computed using inputs averaged to those scales so that comparisons used consistent methods. This creates differences as a function of data resolution. However,

the spreads in Twomey effect and LWP susceptibility estimates from E3SMv1 are of similar magnitude to observational spreads despite being computed at a constant resolution. This occurs because the retrieval assumptions create shifts in the $N_d$ distribution relative to its true shape in direct model output. Retrieval biases are to be expected given assumptions of constant $N_d$ in the cloud layer, drizzle contamination, and scaled adiabatic droplet growth between cloud base and cloud top, but this clearly demonstrates the significant impacts of such assumptions in quantifying aerosol-cloud interactions.

Multiple model-observation comparison approaches like those used in this study coupled with in situ data comparisons provide critical context in the absence of well-quantified sampling and retrieval biases. Comparative analyses used in this study are being

implemented into the open source ESMAC-Diags software package (Tang et al. 2022, 2023) and expanded to additional locations with differing aerosol, cloud, and meteorological conditions such that E3SMv2 and future versions can be assessed in the same manner. However, despite designing this study to have interpretable model-observation comparisons, it does not overcome several key issues that have led to persistent uncertainty of ERFaci through successive generations of climate models. These include the small magnitude of ERFaci relative to the total shortwave cloud radiative effect, its dependency on massive cross-scale atmospheric interactions, an inability to isolate specific model components that contribute to errors, and an inability to estimate such effects with sufficient accuracy in observations (Mülmenstädt and Feingold 2018). Furthermore, it does not overcome the problem of present-day statistically derived cloud susceptibilities being insufficient to describe susceptibilities based on present-day minus pre-industrial conditions (e.g., Ghan et al. 2016, Christensen et al. 2022). There are also observational complexities that were not considered that could impact conclusions. Namely, possible Graciosa Island effects on clouds (e.g., Ghate et al. 2021, Jeong et al. 2022) were not considered. Drizzle contamination can also bias LWP retrievals (e.g., Cadeddu et al. 2020), which can then impact $N_d$ retrievals using LWP as an input, particularly for relatively high LWP and low $N_d$ conditions.

There are several implications of this study's results. First, the diurnal and seasonal cycles of LWP and $N_d$ can modulate cloud albedo susceptibility due to correlations with the SZA, which impacts interpretations of model-observation differences. Second, large uncertainties exist due to $N_d$ retrieval assumptions that limit our ability to accurately quantify factors affecting cloud albedo susceptibility. Continued and expanded integration of complementary in situ, remote sensing, and high-resolution model datasets will aid this effort. Critical to properly predicting albedo susceptibility are improved understanding and quantification of LWP and CF adjustments that respond to poorly simulated small-scale entrainment and precipitation processes operating over days and 100s of kilometers, processes that are not well represented in GCMs. Of equal importance is improving parameterized activation of aerosol particles into cloud droplets, which is known to be problematic (e.g., Ghan et al. 2011) and can turn realistic simulated radiation responses to $N_d$ perturbations into greatly amplified (as in this study) or muted responses to aerosol perturbations. Recent work shows promise in improving aerosol activation parameterization (e.g., Silva et al. 2021), though relationships between $N_d$ and CCN can also be affected by precipitation scavenging that complicates interpretations of observations. Further studies are also required to critically improve simulated drizzle and entrainment processes that influence quantification of both the Twomey effect and cloud adjustments via impacts on cloud adiabaticity. Lastly, more studies are needed to assess how confounding factors such as erroneous responses to meteorology (e.g., Eastman et al. 2021) contribute to susceptibilities and interpretations of model-observation comparisons.

**Code and Data Availability**

Files containing all the observations, E3SM output variables, and necessary E3SM setup files used in this study can be downloaded from https://portal.nersc.gov/project/m3525/avarble/varble_et_al_2023_acp/. Python notebooks used to make these files including retrievals of several variables can also be downloaded from the same location. Original DOE ARM datasets at the ENA site used in this research can be downloaded from https://adc.arm.gov/discovery/#/results/site_code::ena.

**Supplement**

Table S1 and Fig. S1-S13 are available in the Supplement.

## Author Contributions

ACV wrote the manuscript, performed analyses, and created visualizations. PLM performed E3SMv1 simulations. Research conceptualization, and reviewing and editing of the manuscript, was performed by ACV, PLM, MWC, JM, ST, and JF.

## Competing Interests

The contact author has declared that none of the authors has any competing interests.

## Acknowledgements

This research was supported by the "Enabling Aerosol cloud interactions at Global convection-permitting scalES (EAGLES)" project (74358), funded by the U.S. Department of Energy, Office of Science, Office of Biological and Environmental Research, Earth System Model Development program area. This research used resources of the National Energy Research Scientific
Computing Center (NERSC), a U.S. Department of Energy Office of Science User Facility located at Lawrence Berkeley National Laboratory, operated under contract no. DE-AC02-05CH11231 using NERSC awards ALCCERCAP0016315, BER-ERCAP0015329, BER-ERCAP0018473, and BER-ERCAP0020990. We thank the numerous AMR instrument and data mentors for providing data. We also thank Zhibo Zhang and an anonymous reviewer for helpful feedback that improved the quality of this paper. The Pacific Northwest National Laboratory is operated for the U.S. Department of Energy by Battelle Memorial Institute
under contract no. DE-AC05-76RL01830.

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

| Dataset | Short Name | Color and Symbol | Computed Variables | Variables Used from Other Datasets |
|---|---|---|---|---|
| 5-min surface retrievals | Obs Sfc 5min | Black Line and Filled Diamond | Cloud fraction (CF), cloud effective albedo, cloud liquid water path (LWP), cloud layer mean drop number concentration ($N_d$), cloud optical depth (COD), cloud layer mean drop effective radius ($R_{eff}$), cloud base and top heights (CBH, CTH), cloud base and top temperatures (CBT, CTT), cloud depth, surface cloud condensation nuclei (CCN), estimated inversion strength (EIS), 700-hPa relative humidity (RH), cloud adiabaticity, and surface rain rate<br><br>Cloud layer mean drop number concentration ($N_d$) using cloud depth, cloud LWP, and COD in Equation 1 | Obs TOA SZA and cloud IWP |
| 60-min surface retrievals | Obs Sfc 60min | Green Line and Filled Diamond | | |
| 4-km satellite retrievals | Obs TOA 4km | Red Line and Filled Circle | CF (1-deg only), TOA albedo, solar zenith angle (SZA), cloud LWP, COD, near cloud top drop $R_{eff}$, CTH, CTT, and cloud ice water path (IWP)<br><br>Cloud layer mean $N_d$ using TOA-retrieved COD and cloud LWP in Equation 2 | Obs Sfc surface CCN, EIS, and 700-hPa RH |
| 1-deg satellite retrievals | Obs TOA 1deg | Orange Line and Filled Circle | | |
| 1-deg E3SM direct output | E3SM | Blue Fill and Open Square | CF, cloud effective albedo, TOA albedo, SZA, cloud LWP, CBH, CTH, CBT, CTT, cloud depth, surface CCN, EIS, 700-hPa RH, cloud adiabaticity, and surface rain rate<br><br>Cloud layer mean $N_d$ predicted by E3SM | E3SM TOA COSP COD, near cloud top drop $R_{eff}$, and cloud IWP |
| 1-deg E3SM COSP retrievals | E3SM TOA | Green Fill and Open Diamond | COSP CF, COD, near cloud top drop $R_{eff}$, cloud LWP, and cloud IWP<br><br>Cloud layer mean $N_d$ using COSP COD and cloud LWP in Equation 2 | E3SM direct output TOA albedo, SZA, CTT, cloud depth, Surface CCN, EIS, and 700-hPa RH |
| 1-deg E3SM surface retrievals | E3SM Sfc | Orange Fill and Open Circle | Cloud layer mean $N_d$ using E3SM output cloud depth and LWP with E3SM TOA COSP COD in Equation 1 | E3SM direct output cloud effective albedo, SZA, cloud LWP, CTT, cloud depth, surface CCN, EIS, 700-hPa RH, cloud adiabaticity, and surface rain rate<br><br>E3SM TOA COSP COD, near cloud top drop $R_{eff}$, and cloud IWP |

**Table 1:** Each dataset analyzed including its short name and color used in figures along with variables computed by each dataset that are used in comparisons. Inputs used to compute cloud layer mean $N_d$ for each dataset are listed. Variables within each dataset that are obtained from a different dataset are also shown.

| Dataset | Warm, Liquid Cloud Samples | Warm Liquid CF (%) | % Time (CF > 95%) | Samples (CF > 95%) | Samples (CF > 95%, SZA ≤ 65°) | Samples (CF > 95%, SZA ≤ 65°, COD > 4, LWP > 20 g m$^{-2}$) | Samples (CF > 95%, SZA ≤ 65°, COD > 4, LWP > 20 g m$^{-2}$, CCN$_{fit}$ > 0, $\theta_{diff}$ < 2°C) |
|---|---|---|---|---|---|---|---|
| *Obs Sfc 5min* | 190,565 | 70 | 41 | 79,609 | 22,247 | 12,643 | 3,907 |
| *Obs Sfc 60min* | 19,594 | 55 | 17 | 3,243 | 852 | 649 | 197 |
| *Obs TOA 4km* | 17,310 | 75 | 36 | 6,301 | 1,716 | 1,381 | 328 |
| *Obs TOA 1deg* | 19,891 | 59 | 20 | 3,960 | 1,017 | 990 | 224 |
| *E3SM* | 28,224 | 54 | 31 | 8,710 | 2,403 | 1,939 | 1,459 |
| *E3SM Sfc* | 28,224 | 54 | 31 | 8,710 | 2,403 | 1,941 | 1,303 |
| *E3SM TOA* | 21,688 | 48 | 23 | 6,557 | 1,871 | 1,697 | 1,459 |

**Table 2:** Warm (T ≥ 0°C) liquid cloud fraction and sampling as filters are applied for each dataset for only situations with no sub-freezing clouds detected using both IWP and cloud top temperature constraints. For E3SM, IWP is derived from the COSP simulator due to an abundance of very low ice concentrations in the upper troposphere that would remove too many warm, liquid cloud samples. Sensitivity tests indicate that these low ice concentrations that COSP does not detect have little impact on albedo. For non-TOA datasets, multi-layer warm cloud situations have additionally been removed. The Obs TOA 4-km CF is derived from measurements over a 0.5° region.

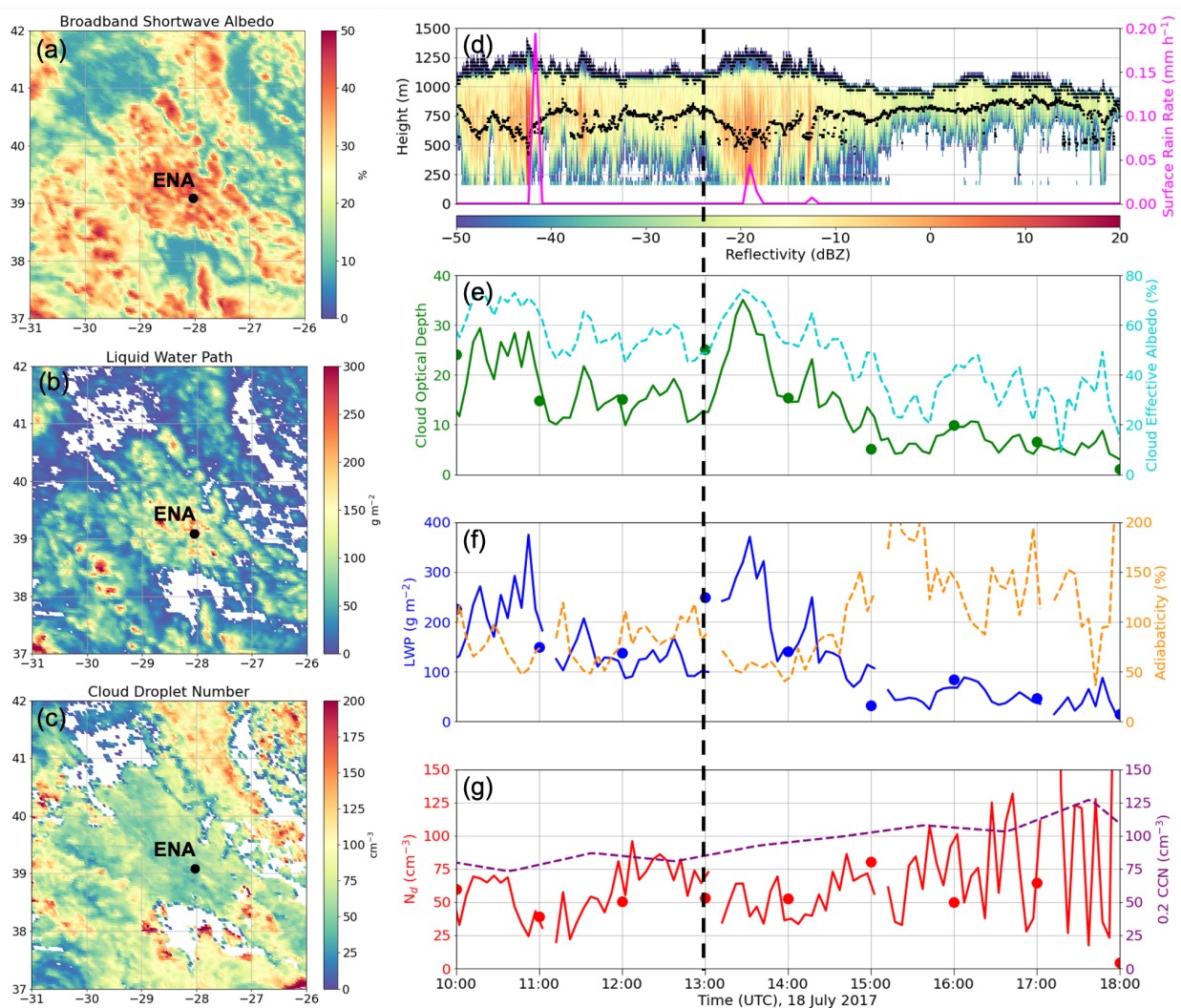

**Figure 1:** An example single layer liquid cloud case at the ARM ENA site showing snapshots of 4-km satellite-retrieved (a) TOA albedo, (b) LWP, and (c) $N_d$ at 1300 UTC 18 July 2017. An 8-hour period of the same event is shown with surface observations of (d) Ka-band reflectivity and lidar-retrieved cloud base with 5-min optical rain gauge surface rain rate, (e) 5-minute surface-retrieved COD (solid) and cloud effective broadband shortwave albedo (dashed), (f) 5-min LWP (solid) and cloud adiabaticity (dashed), and (g) 5-minute $N_d$ (solid) and hourly 0.2% CCN concentration (dashed). Satellite-retrieved COD, LWP, and $N_d$ values are plotted on the time series with circles. The dashed vertical black line indicates the time of the satellite snapshots. The surface site is noted in the satellite images.

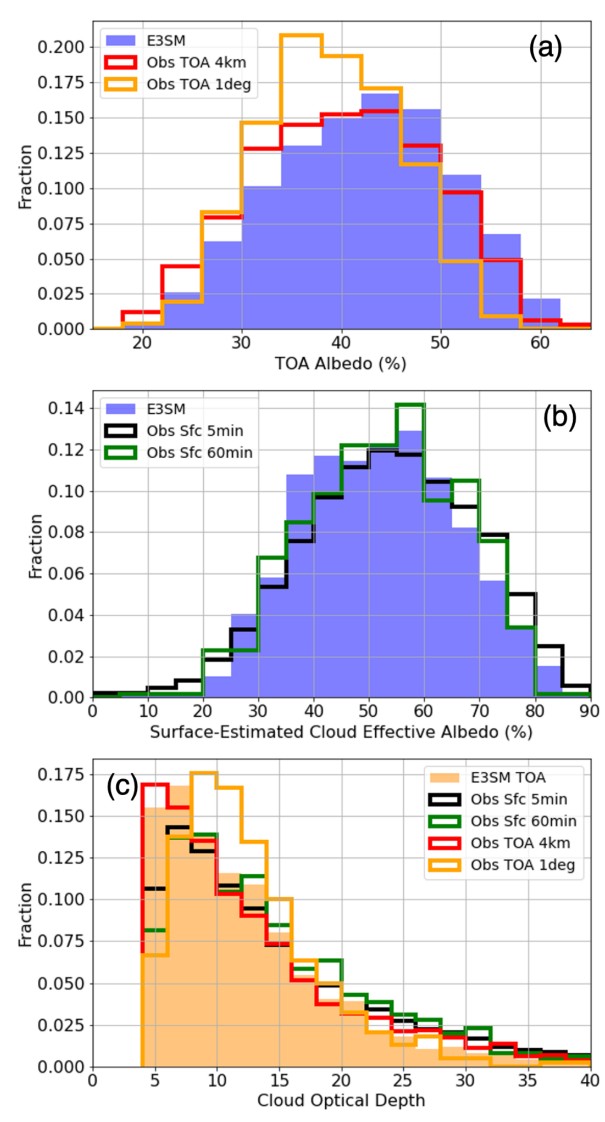

**Figure 2:** Probability distributions of observed and simulated (a) TOA albedo, (b) surface-estimated cloud effective albedo, and (c) cloud optical depth. Datasets are excluded when they are similar to another already shown due to being derived from the shown dataset.

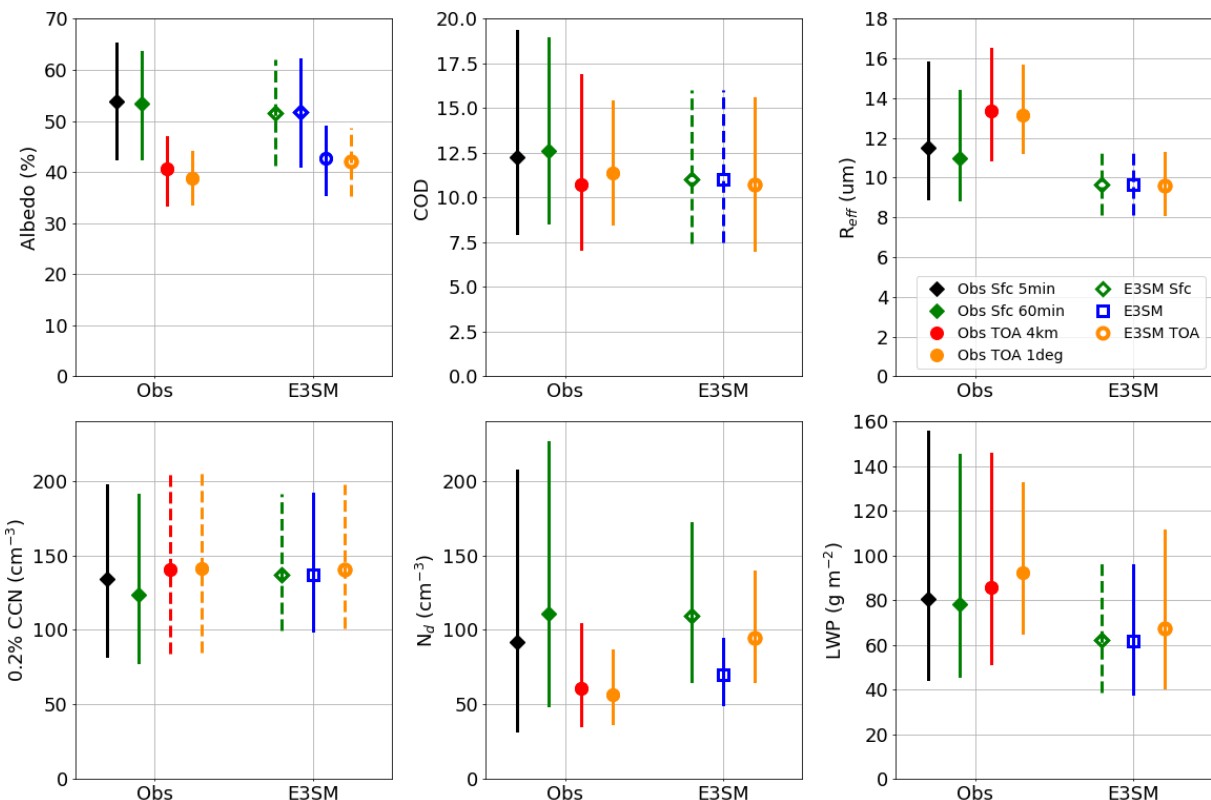

**Figure 3:** Median (symbol) and interquartile range (vertical bar) values of key variables. Dashed vertical bars indicate datasets sampled from solid vertical bar datasets including Obs TOA from Obs Sfc CCN values, E3SM and E3SM Sfc COD and R$_{eff}$ from E3SM TOA, E3SM Sfc and TOA albedos and CCN from E3SM, and E3SM Sfc LWP from E3SM.

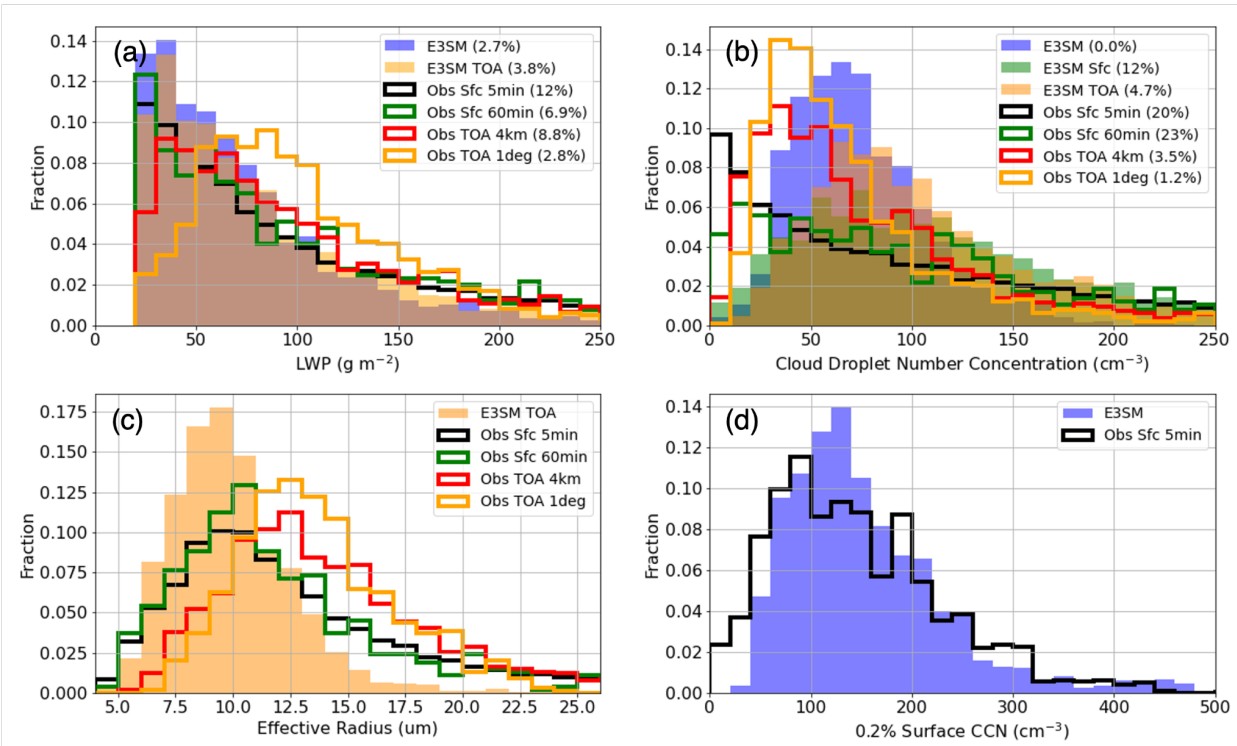

Figure 4: Probability distributions of observed and simulated (a) cloud LWP, (b) cloud layer mean $N_d$, (c) cloud droplet $R_{eff}$ and (d) 0.2% surface CCN. Percentages in the legends indicate how many samples exceed the range of the x-axis. Datasets are excluded when they are similar to another already shown due to being derived from the shown dataset.

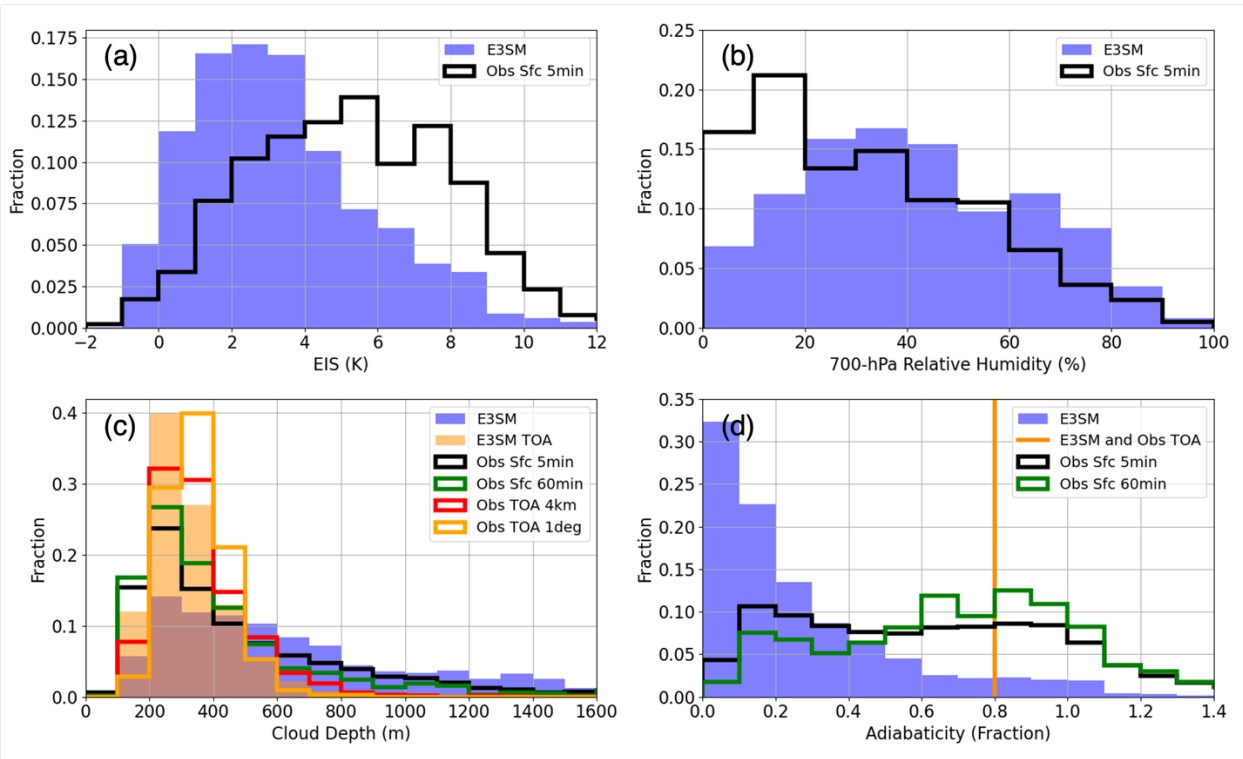

Figure 5: (a) Estimated inversion strength (EIS), (b) 700-hPa RH, (c) cloud depth, and (d) cloud adiabaticity PDFs for E3SM and observations. Observations are derived from interpolated radiosondes. Datasets are excluded when they are similar to another already shown due to being derived from the shown dataset.

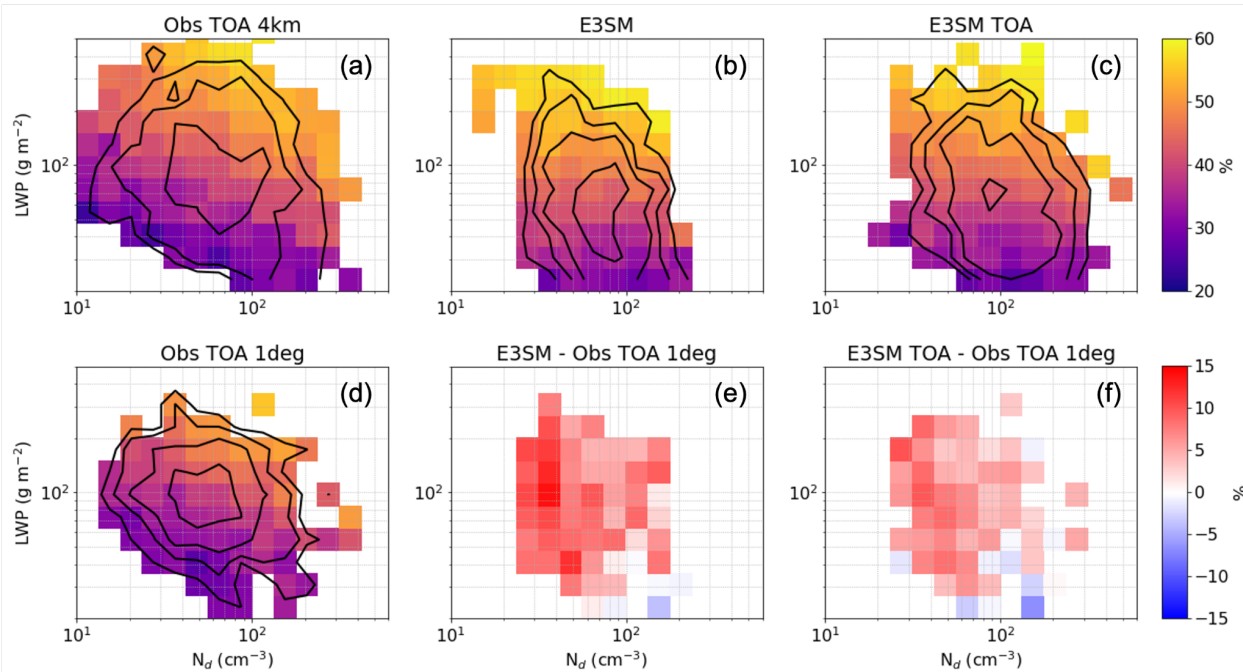

**Figure 6:** Median TOA albedo vs. $N_d$ and LWP for (a) Obs TOA 4km, (b) E3SM, (c) E3SM TOA, and (d) Obs TOA 1deg. Absolute differences are also shown between (e) E3SM and Obs TOA 1deg, and (f) E3SM TOA and Obs TOA 1deg. Black contours indicate sample size thresholds of 0.4, 0.8, 1.6, and 3.2%, respectively.

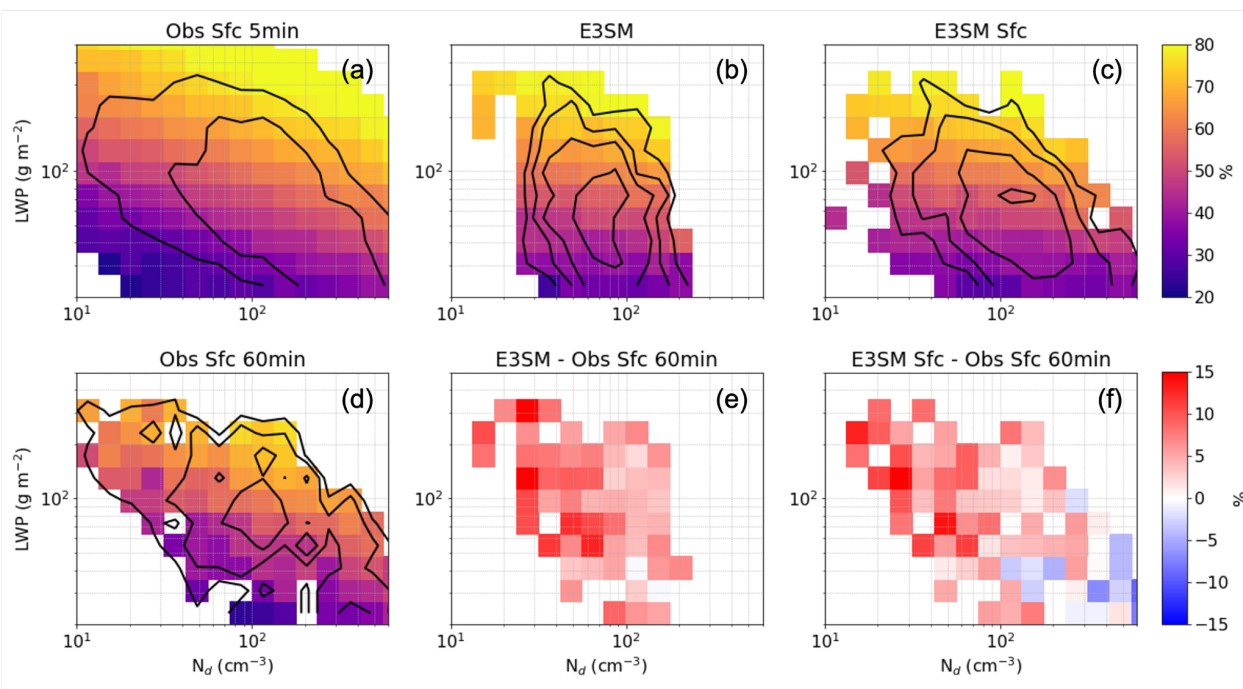

**Figure 7:** Median cloud effective albedo vs. $N_d$ and LWP for (a) Obs Sfc 5min, (b) E3SM, (c) E3SM Sfc, and (d) Obs Sfc 60min. Absolute differences are also shown between (e) E3SM and Obs Sfc 60min, and (f) E3SM Sfc and Obs Sfc 60min. Black contours indicate sample size thresholds of 0.4, 0.8, 1.6, and 3.2%, respectively.

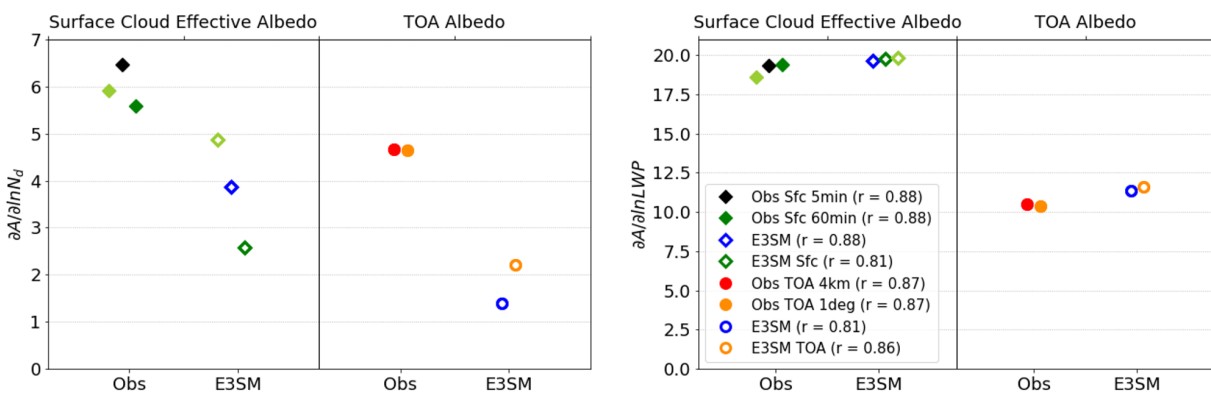

**Figure 8:** Surface cloud effective albedo and TOA albedo as functions of (left) lnN$_d$ and (right) lnLWP for observational and E3SM datasets. Estimates are obtained from ordinary least squares multiple linear regression with Pearson correlation coefficients shown in the legend. Light green symbols represent Obs Sfc 60min and E3SM Sfc datasets with a recomputed N$_d$ that assumes 80% adiabaticity like the TOA retrievals.

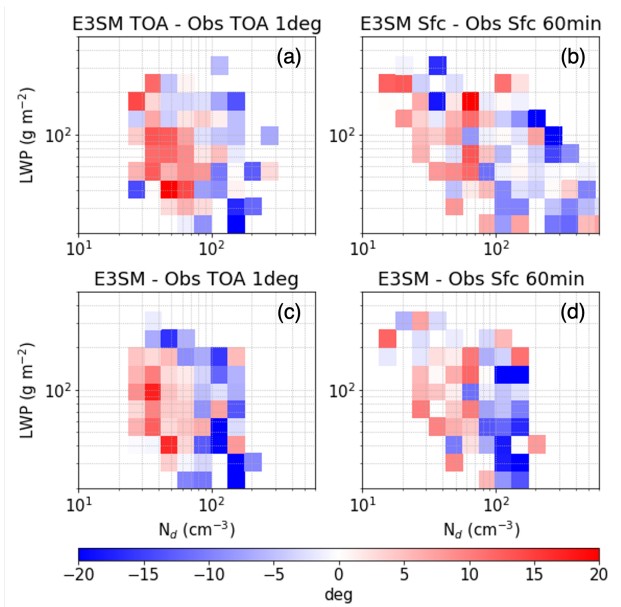

**Figure 9:** Median absolute SZA differences between (a) E3SM TOA and Obs TOA 1deg, (b) E3SM Sfc and Obs Sfc 60min, (c) E3SM and Obs TOA 1deg, and (d) E3SM and Obs Sfc 60min as functions of N$_d$ and LWP.

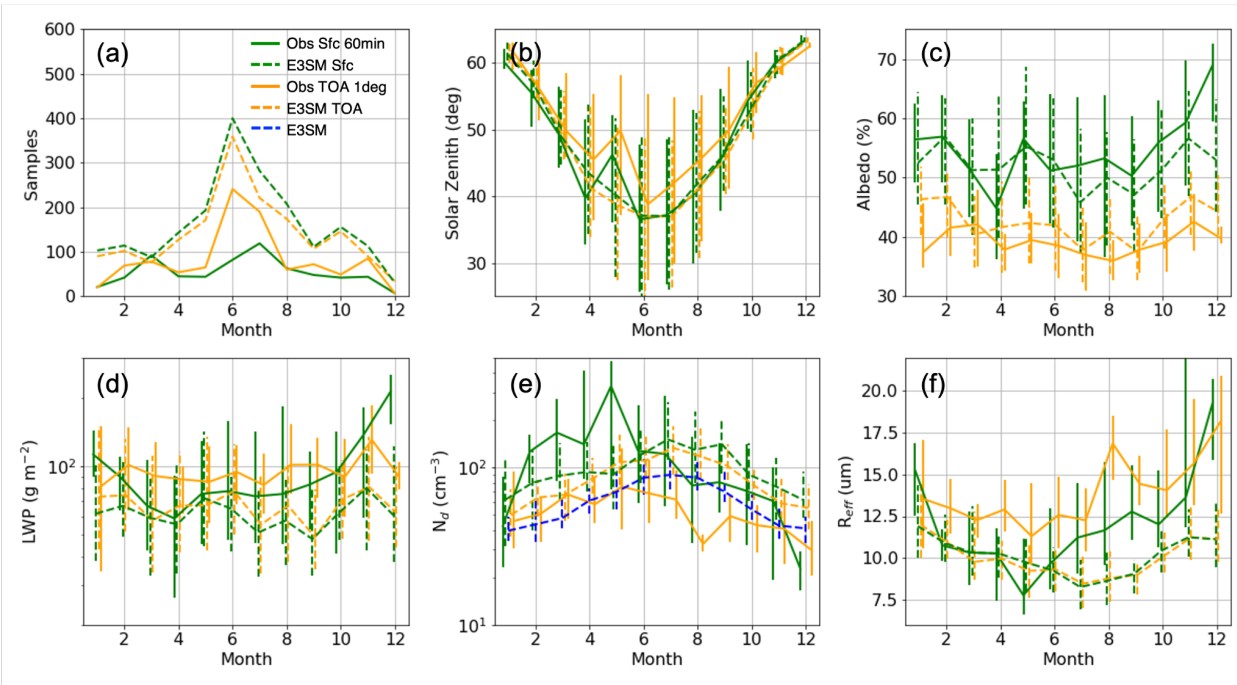

**Figure 10:** Seasonal cycles of (a) number of samples and interquartile ranges (vertical bars) connected by medians for (b) SZA < 65°, (c) TOA or cloud effective albedo, (d) cloud LWP, (e) layer-mean $N_d$, and (f) $R_{eff}$.

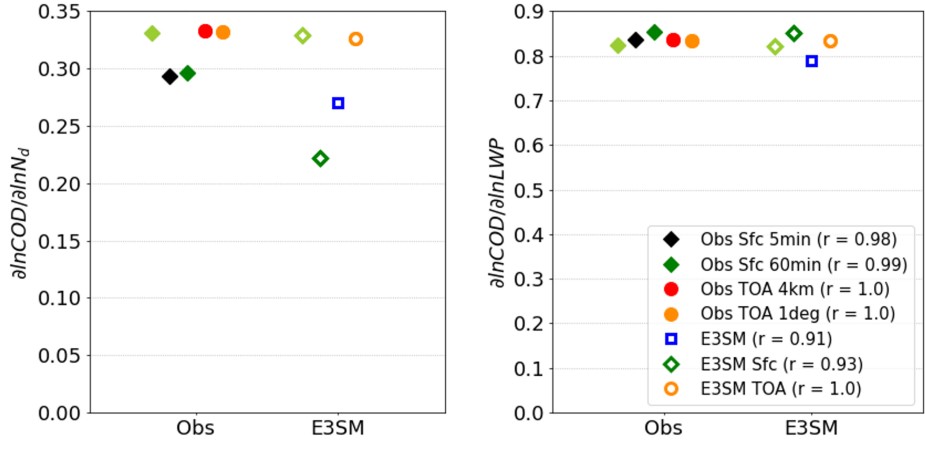

**Figure 11:** lnCOD as a function of (left) $lnN_d$ and (right) lnLWP for observational and E3SM datasets. Estimates are obtained from ordinary least squares multiple linear regression with Pearson correlation coefficients shown in the legend. Light green symbols represent Obs Sfc 60min and E3SM Sfc datasets with a recomputed $N_d$ that assumes 80% adiabaticity like the TOA retrievals.

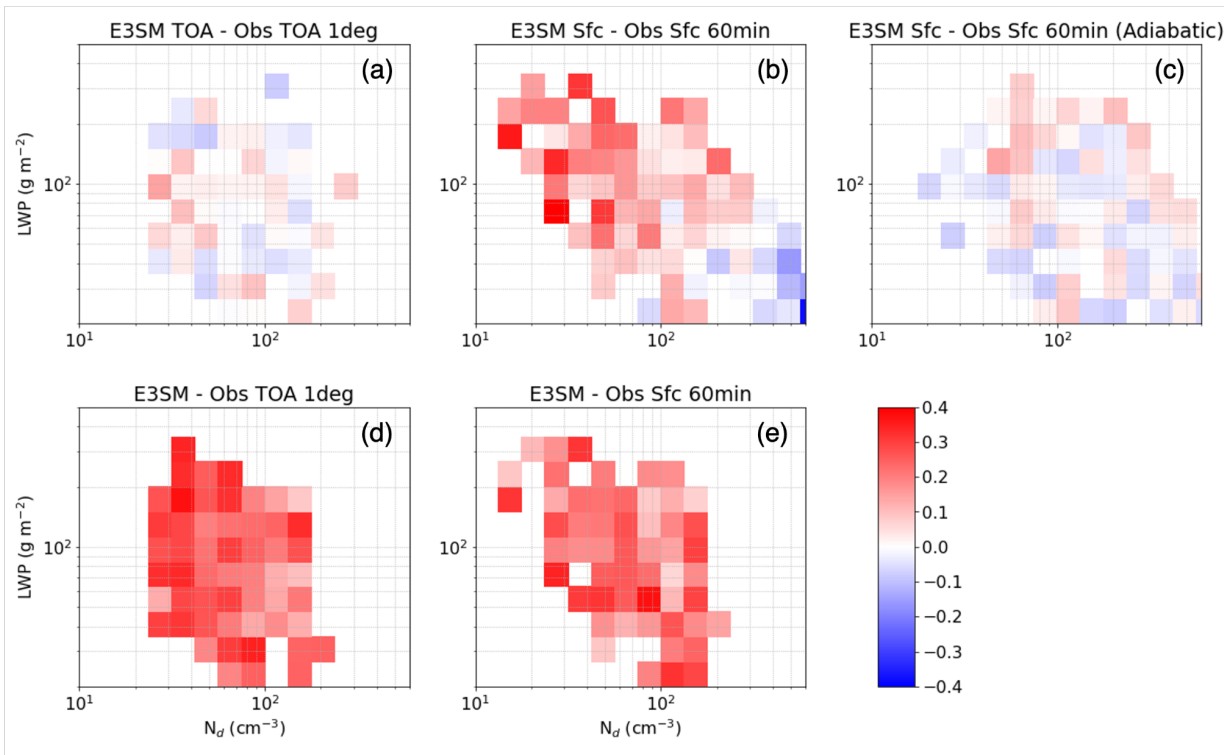

**Figure 12:** Median absolute lnCOD differences between (a) E3SM TOA and Obs TOA 1deg, (b) E3SM Sfc and Obs Sfc 60min, (c) E3SM Sfc and Obs Sfc 60min using $N_d$ retrievals that assume 80% adiabaticity, (d) E3SM and Obs Sfc 60min, and (e) E3SM and Obs TOA 1deg as functions of $N_d$ and LWP.

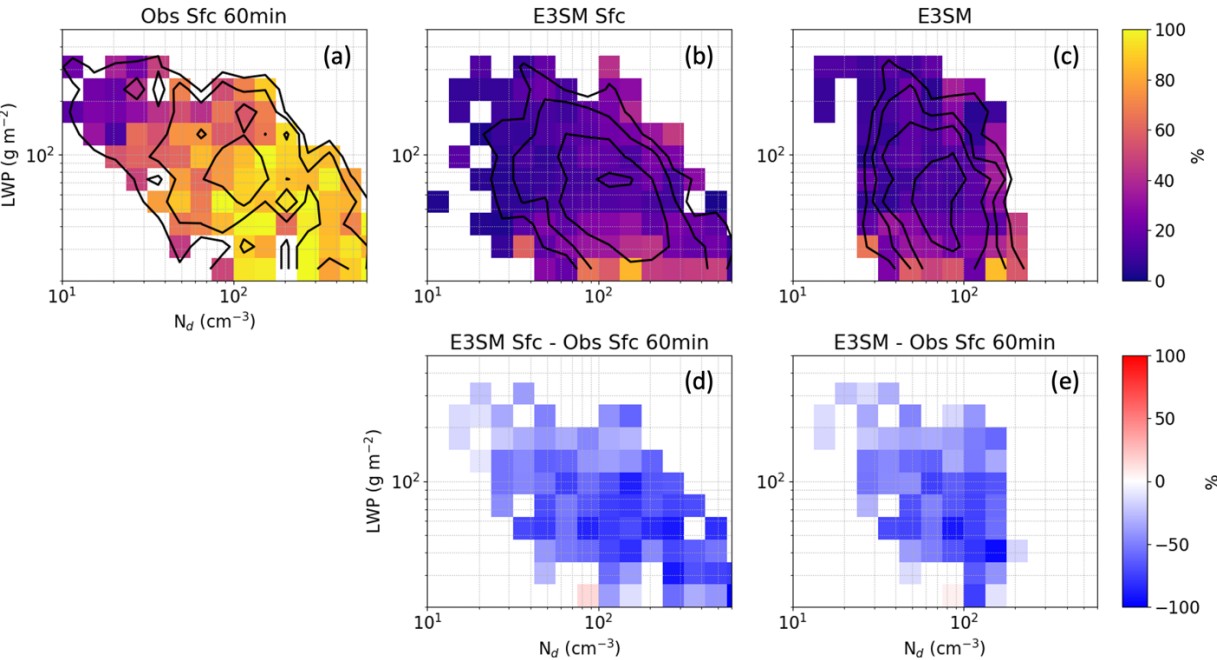

**Figure 13:** Median cloud adiabaticity as a function of LWP and $N_d$ for (a) Obs Sfc 60min, (b) E3SM Sfc, and (c) E3SM. (d) Absolute differences between E3SM Sfc and Obs Sfc 60min. (e) Absolute differences between E3SM and Obs Sfc 60min. Black contours indicate sample size thresholds of 0.4, 0.8, 1.6, and 3.2%, respectively.

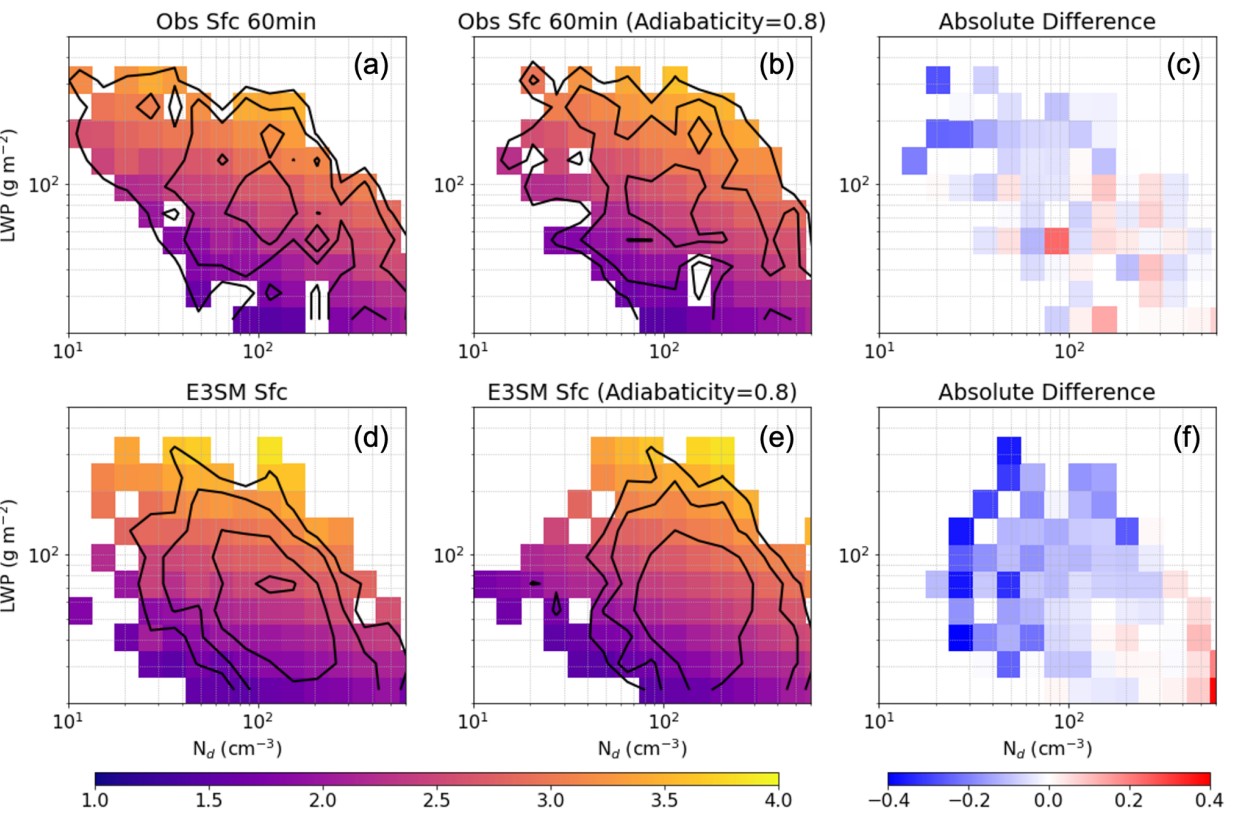

**Figure 14:** Median lnCOD as a function of LWP and $N_d$ for (a) Obs Sfc 60min, (b) Obs Sfc 60min using Nd derived assuming 80% adiabaticity, (c) Obs Sfc 60min (Adiabaticity=0.8) minus Obs Sfc 60min, (d) E3SM Sfc, (e) E3SM Sfc assuming 80% adiabaticity, and (f) E3SM Sfc (Adiabaticity=0.8) minus E3SM Sfc. Black contours indicate sample size thresholds of 0.4, 0.8, 1.6, and 3.2%, respectively.

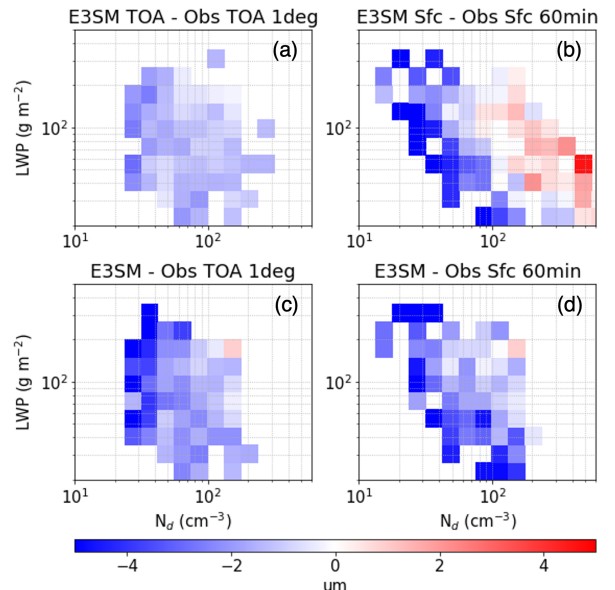

**Figure 15:** Median absolute $R_{eff}$ differences between (a) E3SM TOA and Obs TOA 1deg, (b) E3SM Sfc and Obs Sfc 60min, (c) E3SM and Obs TOA 1deg, and (d) E3SM and Obs Sfc 60min as functions of $N_d$ and LWP.

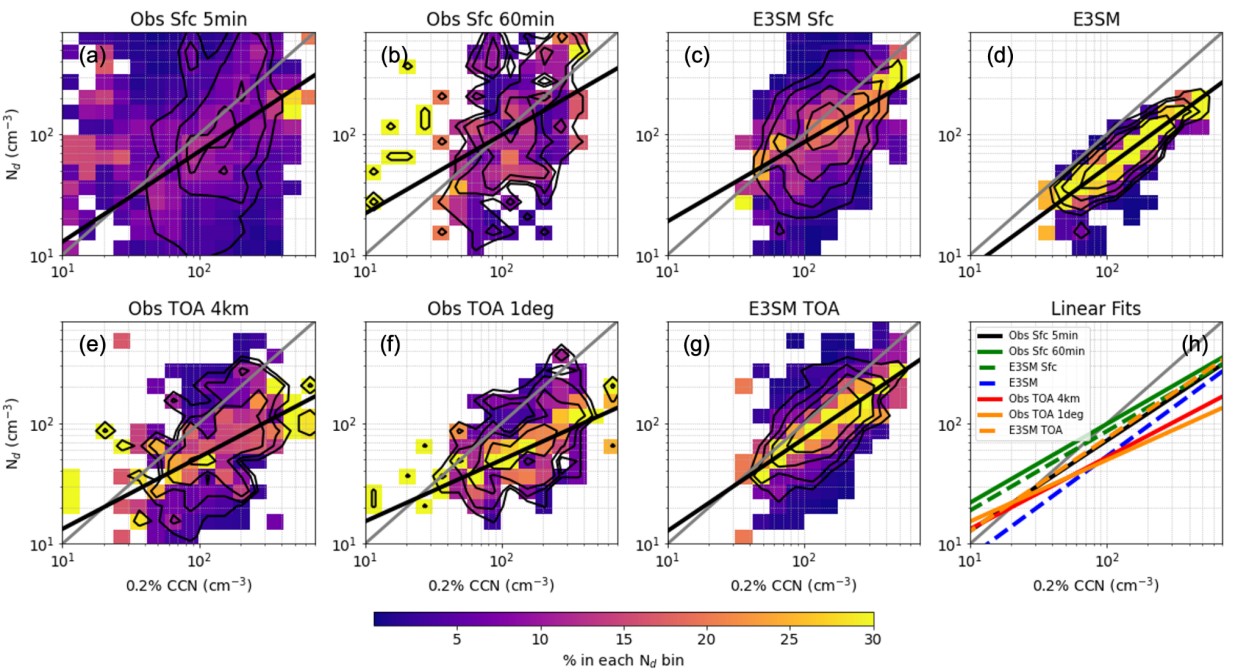

**Figure 16:** Joint distributions of $N_d$ and 0.2% CCN number concentration normalized by CCN bin for (a) Obs Sfc 5min, (b) Obs Sfc 60min, (c) E3SM Sfc, (d) E3SM, (e) Obs TOA 4km, (f) Obs TOA 1deg, and (g) E3SM TOA. Thin black contours indicate sample size thresholds of 0.4, 0.8, 1.6, and 3.2%, respectively. Theil-Sen linear fits are overplotted in thick solid black with the 1:1 line in gray. Linear fits for each dataset are overplotted on one another in (h).

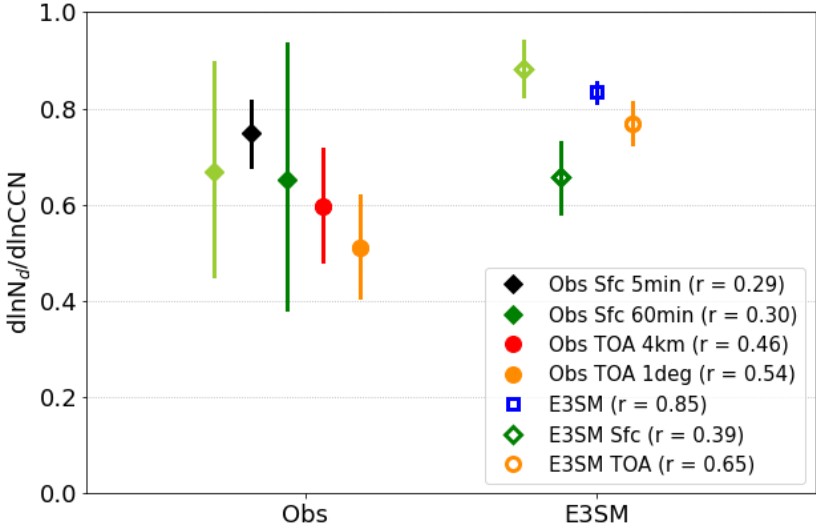

**Figure 17:** $\ln N_d$ as a function of $\ln$CCN for observational and E3SM datasets. Estimates are obtained from Theil-Sen robust linear regression with 95% confidence intervals. Pearson correlation coefficients are shown in the legend. Light green symbols represent Obs Sfc 60min and E3SM Sfc datasets with a recomputed $N_d$ that assumes 80% adiabaticity like the TOA retrievals.

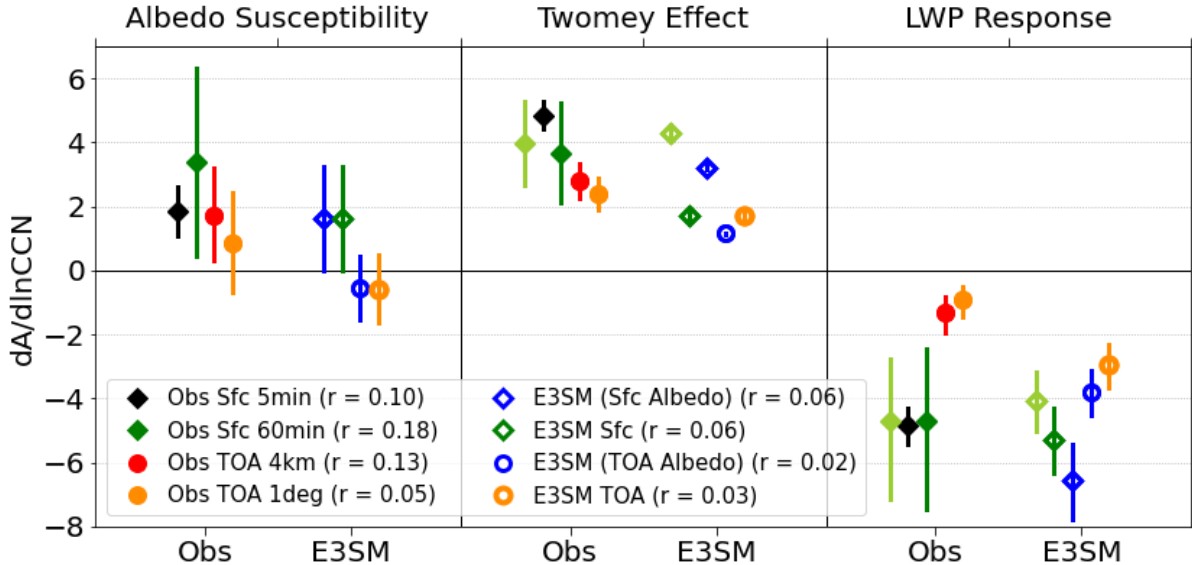

**Figure 18:** (left) Cloud albedo susceptibility to CCN concentration with (middle) Twomey effect $(\frac{\partial A}{\partial lnN_d}\frac{dlnN_d}{dlnCCN})$ and (right) LWP response

$(\frac{\partial A}{\partial lnLWP}\frac{dlnLWP}{dlnN_d}\frac{dlnN_d}{dlnCCN})$ terms for observational and E3SM datasets with 95% confidence intervals. Pearson correlation (r) coefficients for albedo regressed on lnCCN are shown in the legend. Note that confidence intervals and r values can seem inconsistent because of differing sample sizes between datasets (see last column in Table 2). Light green symbols represent Obs Sfc 60min and E3SM Sfc datasets with a recomputed $N_d$ that assumes 80% adiabaticity like the TOA retrievals.

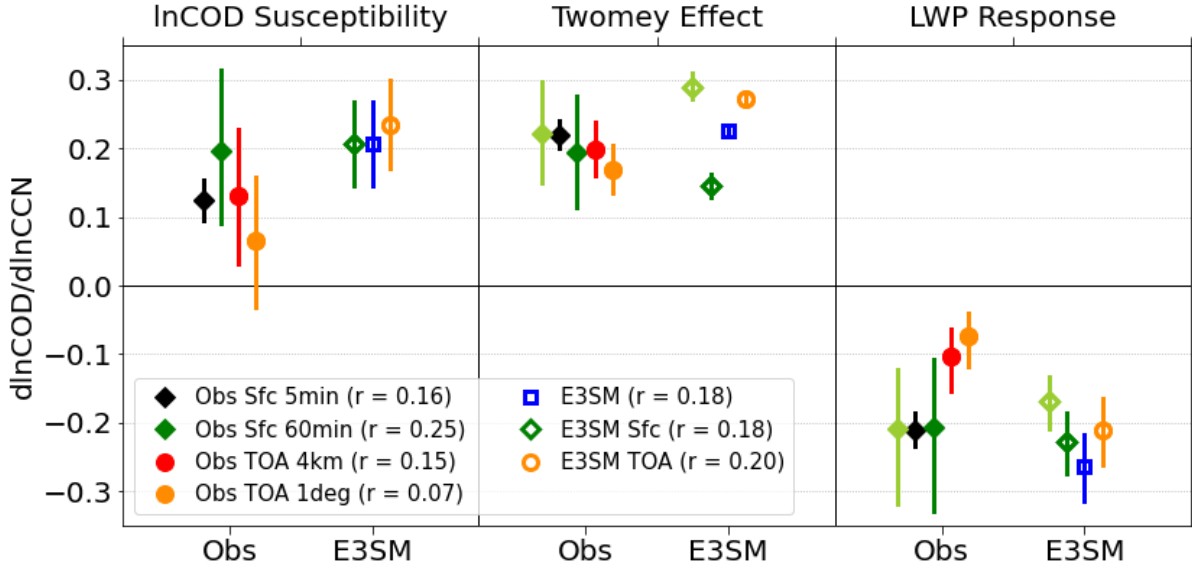

**Figure 19:** (left) lnCOD susceptibility to CCN concentration with (middle) Twomey effect $(\frac{\partial lnCOD}{\partial lnN_d}\frac{dlnN_d}{dlnCCN})$ and (right) LWP response

$(\frac{\partial lnCOD}{\partial lnLWP}\frac{dlnLWP}{dlnN_d}\frac{dlnN_d}{dlnCCN})$ terms for observational and E3SM datasets with 95% confidence intervals. Pearson correlation (r) coefficients for lnCOD regressed on lnCCN are shown in the legend. Light green symbols represent Obs Sfc 60min and E3SM Sfc datasets with a recomputed $N_d$ that assumes 80% adiabaticity like the TOA retrievals.

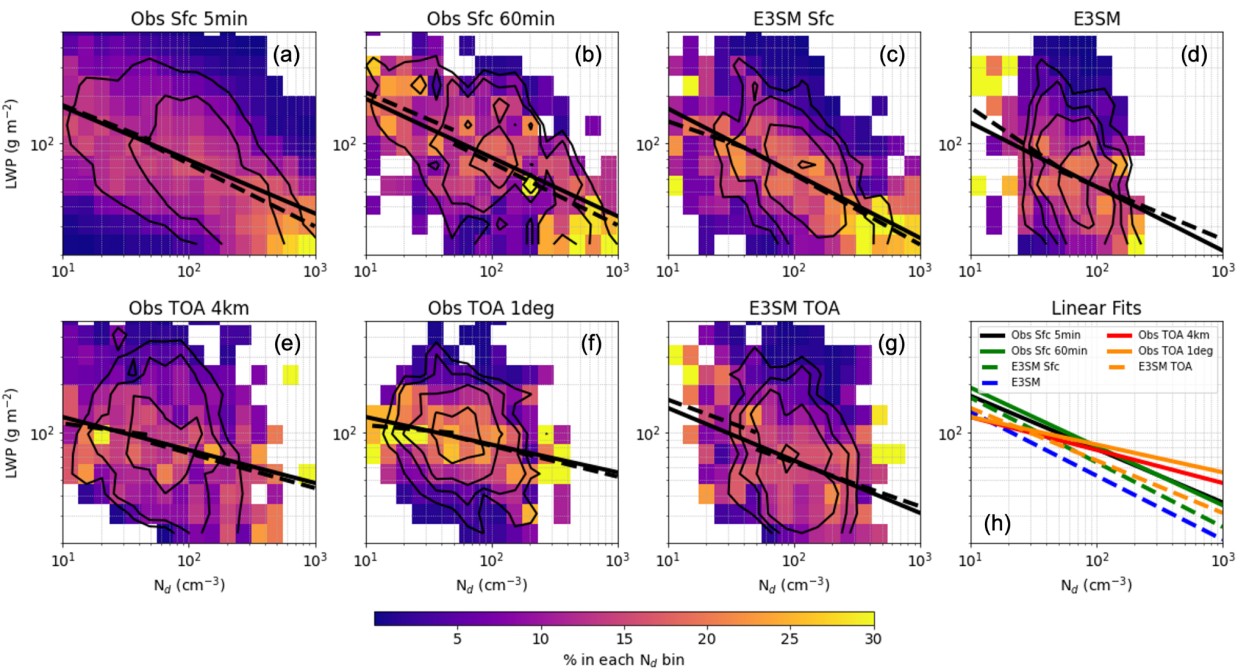

**Figure 20:** Joint distributions of LWP and $N_d$ normalized by $N_d$ bin for (a) Obs Sfc 5min, (b) Obs Sfc 60min, (c) E3SM Sfc, (d) E3SM, (e) Obs TOA 4km, (f) Obs TOA 1deg, and (g) E3SM TOA. Thin black contours indicate sample size thresholds of 0.4, 0.8, 1.6, and 3.2%, respectively. Single Theil-Sen linear fits are overplotted in thick solid black, and piecewise fits are overplotted in thick dashed black. Linear fits for each dataset are overplotted on one another in (h).

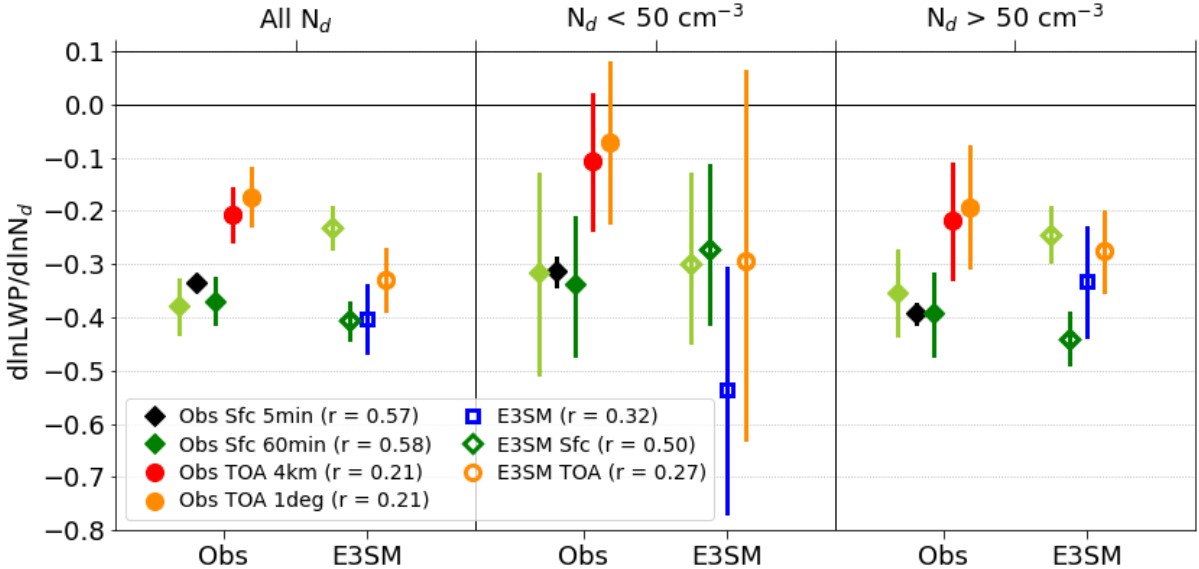

**Figure 21:** lnLWP as a function of ln$N_d$ for all $N_d$ values, $N_d < 50$ cm$^{-3}$, and $N_d > 50$ cm$^{-3}$ for observational and E3SM datasets. Estimates and 95% confidence intervals are obtained from Theil-Sen robust linear regressions with Pearson correlation coefficients shown in the legend. Light green symbols represent Obs Sfc 60min and E3SM Sfc datasets with a recomputed $N_d$ that assumes 80% adiabaticity like the TOA retrievals.

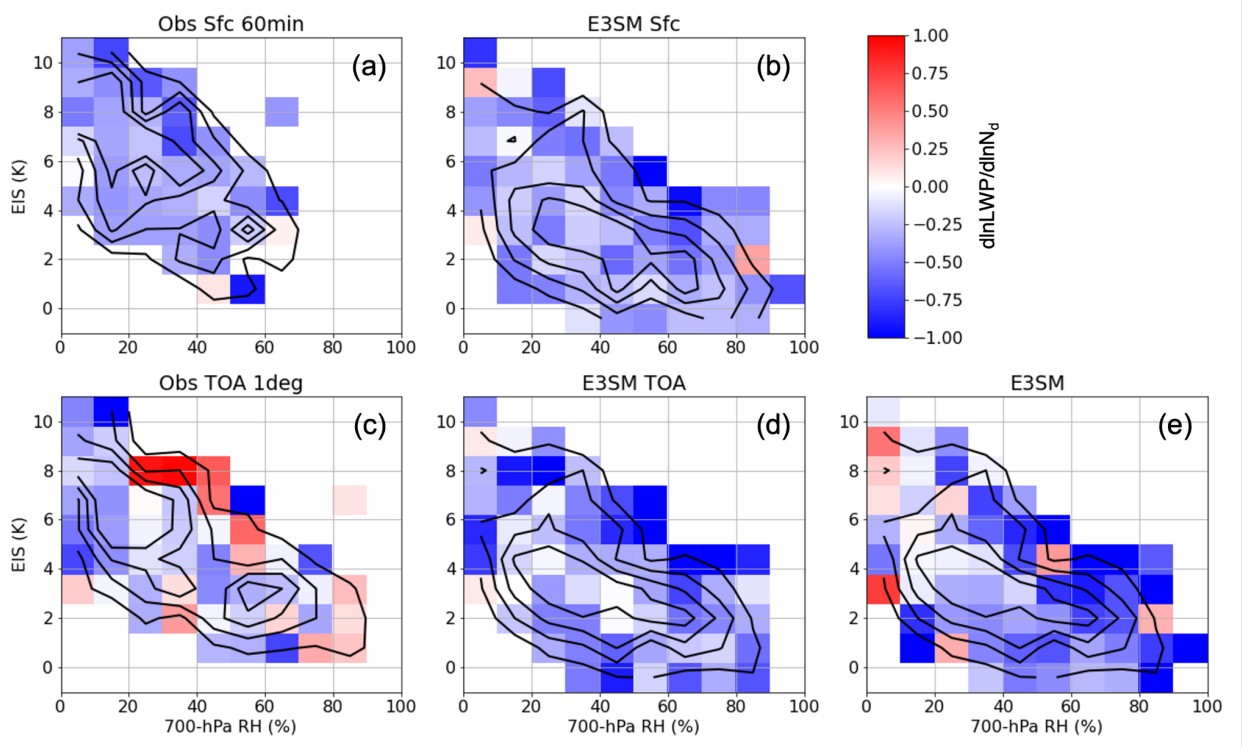

**Figure 22:** Median LWP response ($\frac{dlnLWP}{dlnN_d}$) as a function of EIS and 700-hPa RH for (a) Obs Sfc 60min, (b) E3SM Sfc, (c) Obs TOA 1deg, (d) E3SM TOA, and (e) E3SM. Black contours indicate sample size thresholds of 0.4, 0.8, 1.6, and 3.2%, respectively.

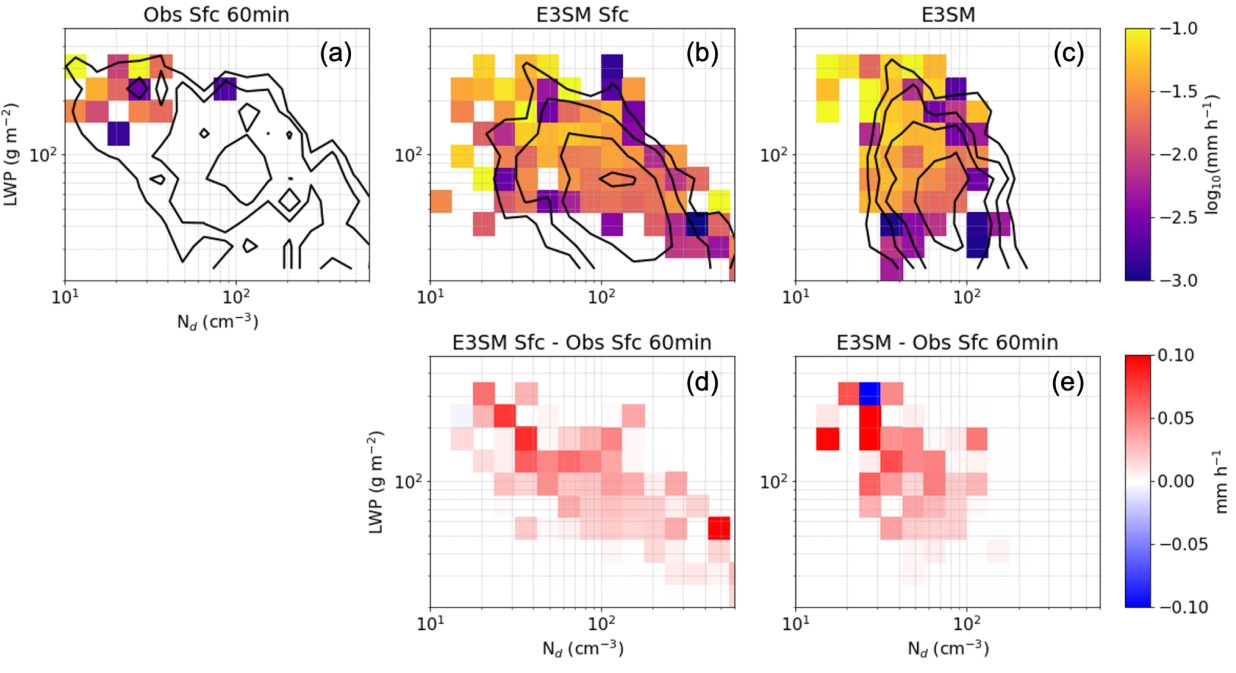

**Figure 23:** Median surface hourly rain rate as a function of LWP and $N_d$ for (a) Obs Sfc 60min disdrometer retrievals, (b) E3SM Sfc, and (c) E3SM. (d) Absolute differences between E3SM Sfc and Obs Sfc 60min. (e) Absolute differences between E3SM and Obs Sfc 60min. Only rain rates exceeding 0.001 mm h$^{-1}$ are included in statistics due to observational sensitivity limitations. Black contours indicate sample size thresholds of 0.4, 0.8, 1.6, and 3.2%, respectively.