# Peer review of "Evaluation of Liquid Cloud Albedo Susceptibility in E3SM Using Coupled Eastern North Atlantic Surface and Satellite Retrievals"

_EGUsphere, 2023_

## Author Comment (AC1)

We thank both reviewers for their time providing very helpful reviews that have improved the quality of the manuscript. Our responses for each comment are in blue below with corresponding paper revision line numbers in red. Line numbers refer to the revised manuscript with tracked changes.

**Reviewer 1**

As indicated by the title, the manuscript reports the evaluation results of albedo susceptibility of marine warm clouds calculated from E3SM simulations. The evaluations make good use of ground- and satellite-based measurements, products, and approaches, and the analyses are comprehensive and thorough. Although in the end, the main sources of the model deficiencies are not surprising and have been pointed out by the literature, the results presented here are important for the users of E3SM. Hence, I recommend it for publication with some minor revisions for clarity and reproducibility.

1) A large focus of the manuscript has been placed on adiabaticity. It would be most useful for readers to see and understand the distribution of model adiabaticity and the observation-model difference in the early section of the manuscript, which will provide much better context for all the tests (E3SM-sfc, E3SM-TOA, etc.) Otherwise, section 2.4 is quite confusing, and the main goal of those tests does not stand out immediately.

Good idea. We have moved the adiabaticity distribution comparison to section 3 and discuss it on lines 318-335. We have also added an earlier discussion of how adiabaticity varies between datasets in section 2.4 on lines 251-260. Some information is included on how the impact of adiabaticity differences is assessed in surface retrieval sensitivity tests where equation 2 is applied with 80% adiabaticity as for TOA retrievals but still using the same COD and LWP inputs as for the surface datasets that make no adiabaticity assumption.

2) Equation 4: It may be useful to decompose further to include adiabaticity explicitly. I would also suggest expanding it for cloud optical depth and effective radius as well. This would help to connect equations, text, and figures in a much more systematic way.

Adiabaticity scales $N_d$ as $\sqrt{\alpha}N_{d,adiabatic}$ and LWP as $\alpha LWP_{adiabatic}$ where $\alpha$ is adiabaticity, thus producing the following equation that we have added as equation 5:

$$\frac{dA}{d\ln CCN} = \left(\frac{\partial A}{\partial \ln\left(\sqrt{\alpha}N_{d,adiabatic}\right)} + \frac{\partial A}{\partial \ln\left(\alpha LWP_{adiabatic}\right)}\frac{d\ln(\alpha LWP_{adiabatic})}{d\ln(\sqrt{\alpha}N_{d,adiabatic})}\right)\frac{d\ln(\sqrt{\alpha}N_{d,adiabatic})}{d\ln CCN}$$

While assuming different constant adiabaticities will affect individual terms, the effects will cancel in the overall albedo susceptibility. The effect of $\alpha$ on albedo could be separated into a third term within the parentheses of equation 4 with a chain rule dependency of $\alpha$ on $N_d$. However, this removes the intuition for how $\alpha$ is affecting sensitivities, which is through its relative effects on adiabatic $N_d$ and adiabatic LWP that are better highlighted by equation 5. Because $\alpha$ varies between 0 and 1, LWP decreases faster than $N_d$ as $\alpha$ decreases, affecting relative sensitivities to $N_d$ and LWP. However, $\alpha$ varies as a function of $N_d$ and LWP, as shown in Figure 13. An empirical fit of $\alpha$ based on $N_d$ and LWP could be performed, but it would differ substantially for observations and E3SMv1, and this would further complicate interpretation of how $\alpha$ is affecting susceptibility terms. Thus, we have stuck with equation 5 in the revised manuscript.

COD in equation 4 would just replace albedo. Alternatively, albedo can be written as a form of COD and solar zenith. $R_{eff}$ is implicit in observational combinations of LWP and COD (which determine $N_d$) with the constant k parameter in equations 1 and 2. In E3SMv1, it is also a diagnostic quantity because cloud and

rain droplet number and mass concentrations are predicted but gamma size distributions are parameterized based on these that set $R_{eff}$ for a given LWC and $N_d$. Thus, one could replace $N_d$ with $R_{eff}$, but having both in equation 4 seems redundant.

We now discuss the above points after equation 4 is introduced on lines 344-359.

3) Please say a few words on how E3SM determines cloud effective radius and compute albedo, and how E3SM simulations differ from COSP calculations. In various tests, has the relationship between liquid water path, cloud optical depth, and cloud effective radius been retained? If not, how does this impact on the interpretation?

We have added information on how $R_{eff}$ and albedo are determined from E3SM output on lines 202-204 and 225-230. $R_{eff}$ and COD profiles are inputs to COSP and are therefore consistent with E3SM output, though COSP gives a value as would be detected by a satellite weighted toward the upper portion of the cloud layer. However, LWP is predicted from COSP based on $R_{eff}$ and COD inputs, whereas LWP for E3SM is predicted and not tied to $R_{eff}$ and COD by a simple equation since size distributions differ between the model and what is assumed in adiabatic cloud retrievals. Thus, the relationship between these variables as well as $N_d$ differs, which is part of the motivation for using multiple retrievals in addition to direct model output for a better interpretable comparison with observational retrievals. This is now discussed on lines 237-245. We have also attempted to clarify which variables are computed within a given dataset versus those that are copied from other datasets in Table 1.

4) ENA site is known to have an island effect. Please say a few words on its impact if the island effect has not been considered and removed in the dataset.

Great point. This information should have been included. We have not removed the island effect from analyses, and this could contribute to differences. We chose not to limit wind directions to those from the northwest through east following Ghate et al. (2021) because this substantially decreases sample sizes, thus increasing sampling uncertainty. In Christensen et al. (2023), the effect of Graciosa Island on climatological cloud properties along Lagrangian trajectories was also not apparent with filtering by Froude number despite obvious effects on specific days. Thus, how the island affects susceptibility terms is unclear. We have added this major caveat on lines 287-291 and lines 633-635 and its investigation is left to future work.

Christensen, M. W., Ma, P.-L., Wu, P., Varble, A. C., Mülmenstädt, J., and Fast, J. D.: Evaluation of aerosol–cloud interactions in E3SM using a Lagrangian framework, Atmos. Chem. Phys., 23, 2789–2812, https://doi.org/10.5194/acp-23-2789-2023, 2023.

5) Min and Harrison (1996) used MWR-based LWP retrieval and flux-based cloud optical depth retrieval to compute effective radius. When the solution is not converging, a default cloud effective radius is used. Please comment on how this dataset has been handled and what is the possible impact if the default situations have not been excluded.

Correct. We have removed times in which the default effective radius is used. We have clarified this in lines 147-148.

6) Do you really mean to cite McComiskey et al. (2009) for Equation (1)?

The citation of McComiskey et al. (2009) was for usage of the specific input datasets. We have clarified this on lines 158-159 and added a sentence on lines 162-163 that the retrieval format follows the adiabatic

cloud model in Bennartz (2007) with incorporation of cloud depth to allow for variable adiabaticity following Boers and Mitchell (1994).

7) Please define cloud effective albedo and check the reference, since Long and Ackerman (2000) talked about cloud radiative effect on downward flux, but not albedo.

We have clarified the definition on lines 154-156.

8) Line 310: I don't understand the logic here. Please elaborate on this a bit.

We have clarified on lines 404-406 that COD susceptibility is analyzed to remove the effects of SZA on correlations between $N_d$ and cloud albedo caused by seasonal correlation of SZA and $N_d$ in E3SMv1 that is not observed. This allows for better comparison of cloud radiative changes in response to $N_d$ and LWP alone with the effects of sun angle. We have also added a note on this on lines 344-346.

9) Line 417: I wouldn't say that they are different. In fact, they cover the same conditions, but the observations cover a wider range. The question is why - do you know?

Good point. We have reworded this on lines 516-517 to state that observations have greater frequency of strong inversions (high EIS) and low 700-hPa RH than simulated. We are not sure why this is the case but have added text in lines 318-323 that the cause for deeper clouds and weaker inversions in E3SMv1 requires further investigation.

10) Line 523: Please be more specific about the forcing you meant here.

We have clarified that this is ERFaci as compared to the total cloud radiative forcing on line 629.

11) Adiabaticity of 200%? Is this mislabeled?

Retrieved adiabaticity greater than 100% can occur due to cloud depth and LWP errors, particularly in low LWP, thin clouds in which LWP errors are ~20 g m$^{-2}$ and cloud depth can be off due to limited radar range gate spacing. We have added this context on lines 326-328.

Reviewer 2: Zhibo Zhang

Review of Evaluation of "Liquid Cloud Albedo Susceptibility in E3SM Using Coupled Eastern North Atlantic Surface and Satellite Retrievals" by Varble et al.

The manuscript presents a comparison study of the susceptibility of liquid-phase clouds in the North Atlantic region (i.e., ENA site) to aerosols perturbations in an Earth System Model (E3SMv1) vs. that derived based on satellite and ground based observations. The study carefully collocated the satellite and ground observations, and used the COSP simulator to ensure that different sources of observations and model simulations are comparable. The comparison reveals several differences in both the mean state and susceptibility of liquid clouds between E3SM simulation and observed counterparts, for example, a greater COD susceptibility to CCN in the E3SM than observation. The potential causes of the differences are analyzed and discussed.

Although the scope of this study is focused/limited, e.g., susceptibility in a particular model in a small region, the methods used in this study and the lessons learned are useful for the user community of E3SM and to the broad community, especially ESM modelers. The manuscript provides in-depth analyses of the model simulations, a bit overwhelming though, and strives to obtain process-level understanding of the potential problems in the model and factors causing observation-model differences. Overall, it is an illuminating paper with useful information, that should be accepted for publication. On the other hand, some parts of the paper are hard to follow, especially for readers not familiar with ESMs. In addition, a few questions, and comments, a listed below, need to be addressed and clarified in the revised paper.

Major comments:

- The motivation and objectives for this study, and its potential significances need to be better explained in the Introduction and echoed in the other parts of the paper. I understand that, to the developers and main users of the E3SM, a comparison study like this is important to understand the performance of the model and the diagnose the potential problems. I also understand that DOE-ARM program has a permanent site in the ENA region which us probably why the study is limited to ENA region. But I guess a significant portion, if not the majority, of the readers are not aware of this background information. So, I would suggest adding some background information to the Introduction about the E3SMv1 model, e.g., why it is an important model, whether it is widely used, why should we care about v1 when other versions are available etc., as well as the significance of the ENA region for aerosol-cloud interactions studies. It will help the readers understand the significance of this study and see potential relevance of this paper to their own research.

  Great point. We have added more background on why E3SMv1 was used on lines 195-197, namely that v2 was only recently released such that v1 is better characterized in many studies and was part of CMIP6. We have also added information on lines 126-137 about why the ENA site was chosen for analyses, namely because of its unique set of long-term aerosol, cloud, and radiation measurements in an important marine cloud regime.

- It seems that the assumption of cloud adiabaticity is an important factor causing the observation-model differences. But I don't think I understand the method and logic for scaling the surface retrievals based on 80% adiabaticity assumption in section 4.1.1 (to me this section is quite confusing). If my understanding is correct, the cloud adiabaticity is diagnosed from the adiabatic cloud LWP (which is estimated based on cloud base T and P and cloud thickness) and observed "true" LWP. As such, the surface-based estimate of Nd can account for the variability of cloud adiabaticity, therefore arguably more accurate. How is it justified to scale a more accurate

measurement based on a simple assumption? Wouldn't a scaling of satellite observations based on surface estimate of adiabaticity be more reasonable? Moreover, the cloud base, top and depth retrievals based on ground based lidar and radar should be pretty accurate. I don't understand what the basis is for scaling the cloud depth in the computation of Nd. To me a reasonable pathway for "apple-to-apple" comparison is to derive an adiabaticity climatology (e.g., monthly mean) or parameterization scheme that can be used to scale satellite observations.

We agree. The surface-based method is likely more accurate because it does not require an adiabaticity assumption due to its usage of cloud depth as a third input alongside COD and LWP to the $N_d$ retrieval. We have tried to clarify this earlier in the manuscript in lines 183-184.

We also agree that it would be ideal to have a parameterization of adiabaticity that could be applied to satellite retrievals, but this would require more work to fine tune and test inputs than is possible in this study. It would benefit from more precise estimates of LWP such as removing drizzle effects over a wider range of conditions than we sampled. The point of applying a constant adiabaticity to the surface retrieved COD and LWP was to assess how much of the difference between surface and TOA retrievals was a result of how adiabaticity is handled (constant at 0.8 in TOA retrievals vs. variable in surface retrievals) as compared to differing COD and LWP inputs. We have tried to explain this more clearly now on lines 251-260 and 377-378.

- It is also not clear to me how model simulations are scaled based on 80% adiabaticity assumption. As discussed later in the paper, E3SM severely underestimates cloud adiabaticity. But once the adiabaticity is changed, it seems to affect everything, not only Nd, but also LWP to COD to Reff and to cloud albedo. Are they all adjusted based on 80% adiabaticity and if so, how? Otherwise, if only Nd is adjusted, then it adjected Nd is no longer physically consistent with other quantalities right? Overall, I'm confused by the objective for adiabaticity-based scaling. Is it intended to demonstrate retrieval uncertainty or model uncertainty?

These sensitivity tests are meant to demonstrate that a portion of the differences between surface and TOA retrieved relationships is due to the adiabaticity assumption. Thus, this is meant to inform retrieval uncertainty, but also model-observation comparisons. This is done by applying the TOA retrieval (equation 2) to COD and LWP inputs used in the surface datasets, which we now more clearly explain in lines 251-260. Thus, only $N_d$ changes. However, $N_d$ changes by differing magnitudes depending on how the retrieved, variable adiabaticity varies with $N_d$ and LWP, which is shown in Figure 13. Because of this, shifts in derivatives involving $N_d$ are not reproducible by assuming any constant value for adiabaticity. How adiabaticity varies with $N_d$ and LWP matters. This is discussed on lines 415-420. $lnN_d$ is also used in both Twomey and LWP responses, so changes in its distribution directly impacts those terms.

- It seems from Figure 4 that the LWP statistics is simulated reasonably well in the model. However, the discussion in Section 4.2.1 suggests the cloud adiabaticity is underestimated. Does this mean that the clouds are too physically thick in the model, or the cloud base heights are biased? Is it possible to compare the model simulated cloud base and thickness with, for example, ground observations? Sorry if I missed it.

Great suggestion. Yes, we did that comparison, but it is easy to miss in the many different analyses. We have moved the adiabaticity and cloud depth distributions up to section 3 with discussion on lines 318-335, so that this is clearer before moving into the more complex multivariate analyses.

- In Figure 17, the overall cloud albedo susceptibility to CCN (dA/dlnCCN) is positive according to observations (both satellite and surface) whereas it is negative in the model if E3SM is sampled using TOA perspective. This seems to be an interesting and potentially important results. But there isn't much discussion about it. What is the cause for that (too negative LWP responses in the model?)?

  The negative values of albedo susceptibility in E3SMv1 are caused by TOA albedo being strongly influenced by solar zenith, which correlates seasonally with $N_d$ at ENA to suppress the Twomey effect. When COD is used in Figure 18, the negative susceptibility disappears. This is discussed in section 4.1.2, and we have added some clarification on lines 478-480.

- Another thing I failed to understand about Figure 17 is that the uncertainty bar (95% confidence interval) is larger in the observation than the model, but the correlation coefficient (r-value) is significantly smaller in the model than observation. Are the two related and results self-consistent?

  Thanks for pointing this out. While the Pearson correlation coefficient is describing the strength of the linear correlation, the uncertainty in the slope of the relationship is also affect by the sample size. Differences in sample sizes between the datasets tend to dominate the uncertainty (95% confidence interval) in relationships quantified by the linear fit slopes. 60-min observations making usage of CCN measurements have the lowest sample size, which is why that dataset has the greatest uncertainties. The differences in sampling are highlighted in Table 2, which we now refer to in the caption of Figure 18 (old Figure 17) in explaining the differences in confidence intervals.

- The exact meaning of "Twomey effect" is vague and sometimes confusing in the paper (e.g., Figure 17 and 18). Does it simply mean . I would suggest using equations to be clear.

  We state on line 361 that the Twomey effect is $\frac{\partial A}{\partial lnN_d}\frac{dlnN_d}{dlnCCN}$, i.e., the albedo change due to CCN effects on $N_d$. However, we agree that this should be restated when it is discussed later in the paper as well. Thus, we have clarified this definition on line 476, 481, and 565. We also added it along with the definition of the LWP response to the captions for Figures 18-19 (old Figures 17-18).

Minor comments:

- Around line 90: Zhang et al. (2022) used the joint PDF of LWP and Nd to understand the microphysical control and spatial-temporal variability of warm rain probability. It should be cited here along with other recent studies.

  Thank you for pointing this out. We have added this citation on line 90.

- Line 140: "why" should be "what"

  We corrected this typo to be "which" on line 155.

- Line 396: Some caveat should be noted when claiming that the simulation of negative LWP response in the E3SM is a "success" because the discussion later seems to suggest that to be a result of error cancelation (i.e., right for the wrong reason).

  Agreed. We have altered the wording to say that E3SMv1's result differs from most GCMs that produce a positive LWP susceptibility on line 493.

- Finally, I applaud and appreciate the effort of the authors to make all the data and scripts used in this study publicly available (see Code and Data Availability session).

Thank you.

Reference:

Zhang, Z., Oreopoulos, L., Lebsock, M. D., Mechem, D. B., & Covert, J. (2022). Understanding the microphysical control and spatial-temporal variability of warm rain probability using CloudSat and MODIS observations. Geophysical Research Letters, 49, e2022GL098863

---

## Author Response (AR2)

**Editor's Review**

The authors have adequately addressed the reviewer's comments. Please also consider including this citation near line 287 with regards to island effects on the MBL at Graciosa:

Jeong, J.-H., Witte, M. K., Glenn, I. B., Smalley, M., Lebsock, M. D., Lamer, K., & Zhu, Z. (2022). Distinct dynamical and structural properties of marine stratocumulus and shallow cumulus clouds in the Eastern North Atlantic. Journal of Geophysical Research: Atmospheres, 127, e2022JD037021. https://doi.org/10.1029/2022JD037021.

Thank you for pointing out this omission. We have added this citation to the latest tracked changes version of the manuscript on lines 281-283 and lines 613-614. It has also been added to the references list.